

# Heterogeneous uptake of amines onto kaolinite in the

# temperature range of 232-300 K

Y. Liu[1,2,3], Y. Ge[1,3], H. He[1,2,3*]

1. State Key Joint Laboratory of Environment Simulation and Pollution

Control, Research Center for Eco-Environmental Sciences, Chinese Academy

of Sciences, Beijing, 100085, China

2. Center for Excellence in Urban Atmospheric Environment, Institute of Urban

Environment, Chinese Academy of Sciences, Xiamen 361021, China.

3. University of Chinese Academy of Sciences, Beijing, 100049, China

---

* Correspondence to: H. He (honghe@rcees.ac.cn)



**Abstract:**
Amines contribute to atmospheric reactive nitrogen ($N_r$) deposition, new particle
formation and the growth of nano- and sub-micron particles. Heterogeneous uptake of
amines by ammonium compounds and organic aerosols has been recognized as an
important source of particulate organic nitrogen. However, the role of mineral dust in
the chemical cycle of amines is unknown because the corresponding reaction kinetics
are unavailable. In this study, the heterogeneous uptake of methylamine (MA),
dimethylamine (DMA) and trimethylamine (TMA) by kaolinite was investigated in the
temperature range of 232-300 K using a Knudsen cell reactor. Lewis acid sites on
kaolinite were identified as dominant contributors to the uptake of amines, utilizing
Fourier transform infrared spectroscopy. The uptake coefficients ($\gamma$) were derived from
the mass accommodation coefficients based on the temperature dependence of the $\gamma$.
The initial effective uptake coefficients ($\gamma_{eff}$) were $(2.27 \pm 0.26) \times 10^{-3}$, $(1.71 \pm 0.26) \times 10^{-3}$
and $(2.95 \pm 0.63) \times 10^{-3}$, respectively, for MA, DMA and TMA on kaolinite at 300 K,
while they increased ~3-fold with decreasing temperature from 300 K to 232 K. The
adsorption enthalpies ($\Delta H_{obs}$) of MA, DMA and TMA on kaolinite were -7.8 $\pm 0.8$, -9.9
$\pm 2.9$ and -9.4 $\pm 1.0$ kJ mol$^{-1}$, respectively, and the corresponding entropy values ($\Delta S_{obs}$)
were -77.1 $\pm 3.2$, -84.1 $\pm 11.8$ and -80.6 $\pm 3.7$ J K$^{-1}$ mol$^{-1}$. The lifetimes of MA, DMA and
TMA attributable to heterogeneous uptake by mineral dust were estimated to be 7.2,
11.5 and 7.7 h, respectively. These values were comparable to the lifetimes of amines
consumed by OH oxidation. Our results reveal that uptake by mineral dust should be
considered in models simulating the chemical cycle of amines in the atmosphere. The
results will also aid in understanding the possible impacts of amines on human health,
air quality, and climate effects.





## 1.0 Introduction


Over the past 200 years, atmospheric emissions of nitrogen-containing compounds

have increased significantly due to increased anthropological emissions from intensive
fossil-fuel combustion, agricultural activity and animal husbandry (Keene et al., 2002).
This has amplified atmospheric reactive nitrogen ($N_r$) concentrations and increased
atmospheric $N_r$ deposition rates (Meunier et al., 2016). These elevated $N_r$ fluxes may
significantly perturb terrestrial, aquatic, estuarine and coastal marine ecosystems (Liu
et al., 2013;van Breemen, 2002). However, reliable predictive capabilities for the
impacts of $N_r$ on ecosystem require the accurate quantification of the deposition fluxes
(Keene et al., 2002) as well as the transformation of atmospheric $N_r$. Amines, whose
atmospheric concentrations are typically 1~14 nmol N m$^{-3}$ and 1-2 orders of magnitude
lower than that of ammonia (Ge et al., 2011;Qiu and Zhang, 2013), are emitted into the
atmosphere from marine organisms, animal husbandry, biomass burning, sewage
treatment, meat cooking, automobiles and industrial processes (Zhang et al., 2012).
Compared with inorganic nitrogen, the budget and the atmospheric chemistry of organic
nitrogen, including amines, are still poorly characterized (Keene et al., 2002;Cornell et
al., 2003).

Amines may contribute 10–20% of the organic content of ambient particles, and

over one hundred different amine species have been frequently observed in the particle
phase (Qiu and Zhang, 2013). It has been recognized that acid-base reactions between
amines and nitric acid (Murphy et al., 2007), sulfuric acid (Almeida et al., 2013;Jen et
al., 2014;Bzdek et al., 2011;Wang et al., 2010a), and methanesulfonic acid (MSA)
(Nishino et al., 2014) contribute to nucleation (Smith et al., 2010;Almeida et al., 2013),
and growth of nano- and sub-micron particles (Qiu and Zhang, 2013). Heterogeneous
uptake of amines onto $(NH_4)_2SO_4$ (Bzdek et al., 2010a;Qiu et al., 2011;Chan and Chan,





2012), $NH_4HSO_4$ (Qiu et al., 2011;Liu et al., 2012a;Chan and Chan, 2012), $NH_4NO_3$
(Liu et al., 2012a;Chan and Chan, 2012;Lloyd et al., 2009), $NH_4Cl$ (Liu et al.,
2012a;Chan and Chan, 2012), humic acid, and citric acid (Liu et al., 2012b) has been
proposed as another possible explanation for particulate amines observed in ambient
particles. The uptake coefficient ($\gamma$) of amines on ultra-fine nanometer-scale ammonium
particles is close to unity (Bzdek et al., 2010b), whereas it is on the order of $\sim10^{-3}$-$10^{-2}$
on coarse particles (Qiu et al., 2011;Liu et al., 2012a) and fine particles of 20-500 nm
particle diameter (Lloyd et al., 2009). The $\gamma$ of amines on citric acid and humic acid is
on the order of $\sim10^{-3}$ and $\sim10^{-5}$ (Liu et al., 2012b), respectively. Similar to the reaction
between $NH_3$ and secondary organic aerosol (SOA) (Liu et al., 2015b;Nguyen et al.,
2013;Lee et al., 2013b) or between $(NH_4)_2SO_4$ and glyoxal (Galloway et al.,
2009;Trainic et al., 2011;Yu et al., 2011;Lee et al., 2013a), carbonyl groups in organic
aerosol can also take up primary and secondary amines to form imine and enamine
compounds in both bulk solution and the particle phase (Zarzana et al., 2012;De Haan
et al., 2009). This may explain the high-MW constituents with a large fraction of
carbon–nitrogen bonds observed in ambient particles (Wang et al., 2010b).

Mineral dust, with a global source strength of 1000-3000 Tg year$^{-1}$, is one of the

most important contributors to atmospheric particles and plays an important role as a
reactive surface in the global troposphere (Dentener et al., 1996). The reactive uptake
of $SO_2$, $HNO_3$, $NO_2$, $N_2O_5$, $O_3$ and $HO_2$ on mineral dust has been widely investigated
(Usher et al., 2003) and has been found to significantly influence sulfate (Zheng et al.,
2015), nitrate and $O_3$ formation by affecting trace gas concentrations and the
tropospheric oxidation capacity through surface processes (Zhang and Carmichael,
1999). On the other hand, the temperature in the atmosphere varies with latitude,
longitude, and altitude above the Earth's surface, as well as with season and time of day.





For example, the tropospheric temperature for latitude 40 °N during June decreases from
room temperature (r.t.) to 220 K with an increase in altitude. The temperature at the
tropopause can reach values much lower than 220 K: down to 180 K above the Antarctic
in winter (Smith, 2003). For most atmospheric reactions, the reaction kinetics is
sensitive to temperature. Thus, it is very important to measure the reaction kinetics
(Crowley et al., 2010;Atkinson et al., 2006) and its temperature dependence for trace
gases, including amines, (Qiu and Zhang, 2013) on mineral dust in order to fully
understand the relevant atmospheric chemistry. However, at the present time, the role
of mineral dust in the chemical cycle of amines in the atmosphere is unknown, because
the corresponding temperature-dependent reaction kinetics for these reactions is
unavailable.
In this work, heterogeneous uptake of amines including methylamine (MA),
dimethylamine (DMA) and trimethylamine (TMA) by kaolinite, which has been found
to be a typical mineral dust crystalline phase in atmospheric particles, was investigated
in the temperature range of 232-300 K. The temperature dependence of uptake
coefficients, relevant thermodynamic parameters and associated environmental
implications are discussed. The results of this study will aid in understanding the
atmospheric chemistry of amines.

**2.0 EXPERIMENTAL DETAILS**
**2.1 Uptake experiments.** Uptake of amines was investigated using a Knudsen cell
reactor coupled to a mass spectrometer (KCMS). This system has been described in
detail elsewhere (Liu et al., 2012b;Liu et al., 2012a;Liu et al., 2010a;Liu and He,
2009 ;Liu et al., 2008 ;Liu et al., 2008). Briefly, the KCMS was composed of three
stages with different working pressures. The first one was the Knudsen cell reactor



working at ~$10^{-4}$ Torr. It consisted of a stainless steel chamber with a gas inlet controlled
by a leak valve, an escape aperture whose area was adjustable using an iris diaphragm,
a sample holder attached to the top surface of a circulating fluid bath, and an ion gauge
(BOC Edwards). The sample in the sample holder could be exposed to or isolated from
the reactants by a lid connected to a linear translator. The temperature of the sample
holder was controlled using a thermostat and cryofluid pump (DFY-5/80, Henan Yuhua
laboratory instrument Co, Ltd.) and measured with an embedded Pt resistance
thermometer. The second stage was a transition chamber pumped by a 60 L·s$^{-1}$
turbomolecular pump (BOC Edwards). The working pressure in this stage was ~$10^{-7}$
Torr and monitored with an ion gauge (BOC Edwards). In the third stage, a quadrupole
mass spectrometer (QMS, Hiden HAL 3F PIC) was housed in a vacuum chamber
pumped by a 300 L·s$^{-1}$ turbomolecular pump (Pfeiffer) at a working pressure of ~$10^{-9}$
Torr measured using an ion gauge (BOC Edwards).

Kaolinite powder was dispersed evenly on the Teflon sample holder with ethanol,

heated at 373 K for 2 h after the solvent was evaporated at room temperature, and then
out-gassed at 298 K in the Knudsen cell reactor for 8 h to reach a base pressure of
approximately $5.0 \times 10^{-7}$ Torr. Amine (MA, DMA or TMA) equilibrated with the
corresponding aqueous solution was introduced into the reactor chamber through the
leak valve. The pressure of amines in the reactor was kept at $3.5 \pm 0.2 \times 10^{-5}$ Torr to
ensure free molecular flow in the reactor. The reactor chamber was passivated with
amines while the sample was isolated from the reactant gas by the sample cover, until
a steady state QMS signal was established. Then, the sample was exposed to the amines
for uptake experiments.

The observed uptake coefficients ($\gamma_{obs}$) were calculated with a Knudsen cell

equation (Tabor et al., 1994; Ullerstam et al. 2003; Underwood et al., 2000), namely,



$$\gamma_{obs} = \frac{A_h}{A_g} \cdot \frac{I_0 - I}{I} \quad (1)$$

where $A_h$ is the effective area of the escape aperture (0.88 mm$^2$) during uptake
experiments and measured according to methods reported previously (Liu et al., 2009a,
b and 2010a, b); $A_g$ is the geometric area of the sample holder (326 mm$^2$); and $I_0$ and $I$
are the mass spectral intensities with the sample holder closed and open, respectively.
The reaction kinetics were measured in the temperature range of 232-300 K.
Analytical grade MA (40 % in H$_2$O, Alfa Aesar), DMA (40 % in H$_2$O, Aladdin
Chemistry Co. Ltd) and TMA (28 % in H$_2$O, TCI), ethanol (Sinopharm Chemical
Reagent Co. Ltd) and kaolinite (Aladdin Chemistry Co. Ltd) were used as received. The
specific surface area (N$_2$-BET) of kaolinite was 71 m$^2$ g$^{-1}$, measured using a
Quantachrome Autosorb-1-C instrument.
**2.2 *In situ* infrared spectra measurements**. The surface species during uptake of
amine by kaolinite were monitored using *in situ* attenuated total reflection Fourier
transform infrared spectroscopy (ATR-FTIR). The particles were prepared by
depositing small droplets containing kaolinite suspension onto the ATR crystal (ZnSe)
with an atomizer, followed by purging with 1 L·min$^{-1}$ zero air to obtain dry particles in
the ATR chamber. After the particles were dried, which was monitored by observing
the IR bands of water, amine vapor was introduced into the reactor by a flow of 200
mL·min$^{-1}$ zero air through a water bubbler containing the amine. *In situ* IR spectra were
recorded on a Nicolet 6700 (Thermo Nicolet Instrument Corporation, USA) Fourier
transform infrared (FTIR) spectrometer equipped with an *in situ* attenuated total
reflection chamber and a high sensitivity mercury cadmium telluride (MCT) detector
cooled by liquid N$_2$. The IR spectra were recorded taking the dried kaolinite as reference
during uptake of amine. All spectra reported here were recorded at a resolution of 4 cm$^{-}$
$^1$ for 100 scans.





**3.0 Results and Discussion**

**3.1 Uptake of amines on kaolinite at 300 K.** Figure 1A, B and C show the typical

uptake profiles for MA, DMA and TMA, respectively. Figure 1D, E and F show the

corresponding profiles of the observed uptake coefficients, which will be discussed later.

In these experiments, the sample mass was ~20 mg and the temperature was held at 300

K. The partial pressures of amines in the reactor were around $5.0 \times 10^{-5}$ Torr, which

corresponded to ~60 ppbv of amines in the atmosphere. Both the molecular ion peak

and the largest fragment, i.e. m/z 31 and 30 for MA, m/z 45 and 44 for DMA, and m/z

59 and 58 for the TMA, were scanned to verify that the species were being correctly

measured. As shown in Fig. 1A-C, the relative signals of the two monitored mass

channels for each amine coincided very well. The normalized QMS signals decreased

significantly when the kaolinite samples were exposed to amines. The lowest values of

the normalized QMS signal ($I/I_0$) were 0.45, 0.63, and 0.47 for MA, DMA and TMA on

kaolinite, respectively, under this specific condition. Then, the $I/I_0$ increased gradually

with exposure time due to the surface saturation of amines. The uptake behaviors for

these amines on kaolinite were similar to those on humic acid as observed in our

previous work (Liu et al., 2012b), whereas they were different from the uptake

behaviors of amines on ammonium compounds (Liu et al., 2012a). The recoveries of

the uptake curves of amines can be explained by the saturation of the surface reactive

sites on kaolinite. However, exchange reactions take place between amines and

ammonium compounds, resulting in ammonia as the product (Liu et al., 2012a;Qiu et

al., 2011;Bzdek et al., 2010a); subsequently, a continuous uptake of amines

accompanied by ammonia formation was observed within a certain period of time (Liu

et al., 2012a).

Kaolinite is a 1:1 layer mineral. Each layer of the mineral consists of an alumina



octahedral sheet and a silica tetrahedral sheet that share a common plane of oxygen
atoms (Brindley and Robinson, 1945) and repeating layers of the mineral are hydrogen-
bonded together (Miranda-Trevino1 and Coles, 2003). Usually, both Lewis acid and
Brønsted acid sites are present on metal oxides (Busca, 1999;Benvenutti et al., 1992).
Pyridine, piperidine or *n*-butylamine have been widely used as probe molecules to
measure the type of acid site and the acidity of metal oxide materials (Busca, 1999). To
confirm the reactive sites on kaolinite, infrared spectra were collected using an *in situ*
ATR-FTIR.
Figure 2 shows the ATR-FTIR spectra of MA, DMA and TMA adsorbed on
kaolinite for 30 min at 300 K. It should be pointed out that the absolute intensities for
the bands of surface species resulting from adsorption of different amines were not
identical, even for the adsorbed water bands at ~3400 and ~1640 cm$^{-1}$. This might be
related to different mass loading of kaolinite deposited on the ZnSe crystal via
atomization. Therefore, we only qualitatively discuss the assignments of the related
surface species. The typical IR bands of amines, such as the $v_{as}(CH_3)$ and $v_s(CH_3)$ bands
at 2962 and 2860 cm$^{-1}$ accompanied by the $\delta(CH_3)$ bands at ~1400 cm$^{-1}$ and the $\rho(CH_3)$
bands from 985 cm$^{-1}$ to 1290 cm$^{-1}$ (Murphy et al., 1993), the overtone/combination
bands of –CH$_3$ and CN groups in the range of 2300-2700 cm$^{-1}$ (Murphy et al., 1993),
the $v_s(CN)$ bands at ~1260 and 850 cm$^{-1}$ (Durgaprasad et al., 1971), $\delta(NH_2)$ of MA at
1606 cm$^{-1}$ (Nunes et al., 2005) and the $\delta(NH)$ band (NIST) and/or $\delta(CH_3)$ of DMA at
1473 cm$^{-1}$ (Lin et al., 2014) in Fig. 2, confirmed the adsorption of amines on kaolinite.
In a previous work, the IR bands at 1484 and 984 cm$^{-1}$ were assigned to the
characteristic bands of protonated TMA (($CH_3)_3NH^+$) in an acidic solution mixed with
TMA and on the surface of polyethylene (PE) treated by TMA (Ongwandee et al., 2007).
In this work, two bands at 1477-1467 cm$^{-1}$ and 971 cm$^{-1}$ were observed, as shown in





Fig. 2, and were probably related to the protonated amines (MAH⁺, DMAH⁺ and
TMAH⁺). In particular, a strong peak at 1473 cm$^{-1}$ was observed (Fig. 2B), while the
band was very weak for the spectra in Fig. 2A and C. However, this band was very
close to the δ(NH) band at ~1470 cm$^{-1}$ in DMA (NIST database) and/or δ(CH₃) (Lin et
al., 2014) in amines. On the other hand, it should be noted that formation of protonated
amines requires surface OH (Brønsted acid). However, as shown in Fig. 2, the surface
hydroxyl (-OH) in the range of 3600-3750 cm$^{-1}$ (Miranda-Trevino1 and Coles, 2003)
was not consumed when the kaolinite was exposed to MA, DMA or TMA. This
indicates that the content of reactive Brønsted acid sites on the kaolinite must be very
low even if the bands at 1477-1467 cm$^{-1}$ originated from the protonated amines.
Therefore, Lewis acid sites on the kaolinite predominantly contributed to the uptake of
amines as observed in Figs. 1 and 2 in this study. This was similar to the adsorption of
DMA and N,N-dimethyl formamide (DMF) on TiO₂, for which Lewis acid sites were
identified as the reactive sites (Lin et al., 2014).
As shown in Fig. 1, the saturation times for the amines on kaolinite varied slightly
depending upon the number of substituted methyl groups in the amines. For example,
the saturation time was ~20 min for MA, while it decreased to ~15 min and ~10 min
for DMA and TMA, respectively. This implies a different rate of increase for the surface
coverage among the different amines during adsorption. According to the gas-particle
equilibrium of amines on the surface of kaolinite as shown in Eq. (2),

$$A_g \underset{k_d}{\overset{k_a}{\rightleftharpoons}} A_s \quad (2)$$


the surface concentration of amines on kaolinite can be described as,
$\qquad \frac{dc_{A,s}}{dt} = k_a c_{A,g} - k_d c_{A,s} \quad (3)$
where $k_a$ and $k_d$ are the adsorption coefficient and desorption coefficient, respectively;





$c_{A,g}$ and $c_{A,s}$ are the concentration of amines in the gas phase and on the surface,
respectively. When replacing $c_{A,s}$ with the surface coverage of amine ($\theta=c_{A,s}/c_T$), the
time dependent surface coverage of amine is
$$\frac{d\theta}{dt} = \frac{k_a c_{A,g}}{c_T} - k_d\theta \quad (4)$$
or,
$$\theta = \frac{k_a c_{A,g}}{k_d c_T} - \frac{1}{k_d}\exp(-k_d t) \quad (5)$$
where $c_T$ is the total reactive or adsorptive sites or the saturated adsorption capacity for
amines on kaolinite; $t$ is the exposure time. Therefore, the saturation time of amines on
kaolinite depends on the adsorption coefficient, the desorption coefficient, the total
number of reactive sites and the concentration of amines in the gas phase. If all the
layers of the packed kaolinite particles are assumed to be available to amine molecules,
the saturated adsorption capacity of MA, DMA and TMA on kaolinite is estimated to
be $1.7\times10^{19}$, $7.3\times10^{18}$ and $5.7\times10^{18}$ molecules mg$^{-1}$, respectively, under this reaction
condition. These values will be lower if amines cannot penetrate all the layers in the
sample holder. This will be discussed later. The adsorption capacities are inversely
correlated with the cross-sectional area of MA (0.243 nm$^2$), DMA (0.323 nm$^2$), and
TMA (0.394 nm$^2$) (Liu et al., 2012b). This means that a part of the Lewis acid sites
accessible to MA on the kaolinite surface would not be accessible to amines with larger
molecular volume or cross-sectional area.
**3.2 Reaction kinetics of amines on kaolinite at 300 K.** Based on the measured QMS
signals of amines as shown in Fig. 1, the uptake coefficients were calculated using Eq.
(1). Figure 1D, E and F show the evolution of $\gamma_{obs}$ of MA, DMA and TMA, respectively,
as a function of exposure time on kaolinite at 300 K. The initial $\gamma_{obs}$ varied from
~$2.0\times10^{-3}$ to ~$3.0\times10^{-3}$ for these amines at 300 K, while the $\gamma_{obs}$ decreased markedly




with increasing exposure time as shown in Fig. 1. This is in agreement with the results
observed for most of the reactive gases, such as $NO_3$, $N_2O_5$ (Tang et al., 2010), $NO_2$
(Wang et al., 2012;Ndour et al., 2008;Liu et al., 2015a), $O_3$ (Hanisch and Crowley,
2003), HONO (El Zein and Bedjanian, 2012) and COS (Liu et al., 2010a) on typical
atmospheric particles because of surface saturation, as discussed above.

For packed powder samples in a sample holder, diffusion of reactive molecules into

the underlying layers was widely observed in previous works (Liu et al., 2010b;Liu et
al., 2010a;Keyser et al., 1991;Underwood et al., 2001;Grassian, 2002). Therefore, the
effective uptake coefficient ($\gamma_{eff}$) might be overestimated when calculating the $\gamma$ with
Eq. (1) in which the geometric area of the sample holder instead of the effective surface
area is considered. The KML model (Keyser et al., 1991), LMD model (Underwood et
al., 2001;Grassian, 2002) and FPL model (Hoffman et al., 2003) have been developed
to estimate the effective surface area of uptake for heterogeneous reactions. The linear
mass dependent (LMD) model (Underwood et al., 2001), which was developed based
on the KML model, has been widely used for data interpretation in Knudsen cell
experiments. That is,

$\gamma_{obs} = \gamma_{eff} \, m_{eff} \, S_{BET}/A_g$ (6)

or,

$\gamma_{eff} = A_g \, Slope/S_{BET}$   (7)

where $\gamma_{eff}$ is the effective uptake coefficient; $m_{eff}$ is the effective sample mass; $S_{BET}$ is
the specific surface area of the sample; $A_g$ is the geometric area of the sample holder;
*Slope* is the slope of the plot of $\gamma_{obs}$ versus sample mass in the linear regime ($mg^{-1}$).
Thus, $\gamma_{eff}$ can be determined by measuring the $m_{eff}$ of reactive molecules or the slope of
the $\gamma_{obs}$ versus sample mass for multilayer powder samples when the powder samples
evenly cover the sample holder.



Figure 3 shows the $\gamma_{obs}$ of the three amines on kaolinite in the mass range of 20-
100 mg at 300 K. When the sample mass was less than 20 mg, the sample holder could
not be evenly covered by particles. In such a case, $\gamma_{obs}$ would be underestimated with
Eq. (1). Therefore, uptake experiments with sample mass below 20 mg were not
performed. However, as shown in Fig. 3, the $\gamma_{obs}$ of MA, DMA and TMA were
independent of the sample mass of kaolinite within experimental uncertainty. This is
similar to the uptake of amines on $(NH_4)_2SO_4$, $NH_4HSO_4$, $NH_4NO_3$, and $NH_4Cl$ (Qiu et
al., 2011;Liu et al., 2012a) and citric acid (Liu et al., 2012b), the uptake of $HNO_3$ on
soot (Muñoz and Rossi, 2002) and $N_2O_5$ and $H_2O$ on mineral dust (Seisel et al.,
2005;Seisel et al., 2004). This indicates that the underlying layers of the kaolinite
sample contributed very little to amine uptake.
The probe depth of a reactive gas in a packed powder sample can be expressed by
using a factor for the effect of gas-phase diffusion into the underlying layers (Keyser et
al., 1991;Underwood et al., 2001).
$$\eta = \frac{1}{\phi}\tanh(\phi) \quad (8)$$
$$\phi = \frac{m}{\rho_b A_g d_p}\left(\frac{3\rho_b}{\rho_t - \rho_b}\right)\left(\frac{3\tau\gamma_{eff}}{4}\right)^{1/2} \quad (9)$$
Thus,
$$\gamma_{obs} = \gamma_{eff}\,(A_e + \eta A_i)/A_g \quad (10)$$
where $\eta$ is a factor to account for the effect of gas-phase diffusion into the underlying
layers ($0 \leq \eta \leq 1$); $m$ is the sample mass; $\rho_b$ and $\rho_t$ are the bulk density and the true density
of the sample, respectively; $d_p$ is the particle diameter of the sample; $\tau$ is the tortuosity
factor of the sample; $A_g$ is the geometric area of the sample holder; $A_e$ and $A_i$ are the
area of the first layer of the sample (external) and the area of the underlying layers of
the sample (internal), respectively. The reactive gas can effectively diffuse into the



underlying layers if $\eta$ is close to 1, while the contribution of the underlying layers to
the reactive gas uptake is negligible when $\eta$ is close to 0. Therefore, a large $\gamma_{eff}$ should
result in a small $\eta$ or small probe depth, and vice versa. This means that the $\gamma_{eff}$ of
amines on kaolinite should be close to or equal to the $\gamma_{obs}$ in this study. Thus, the mean
$\gamma_{obs}$ measured at different sample mass were taken as the corresponding $\gamma_{eff}$ of amines
on kaolinite and summarized in Table 1. They were $(2.27\pm0.26)\times10^{-3}$, $(1.71\pm0.26)\times10^{-3}$
$^{-3}$ and $(2.95\pm0.63)\times10^{-3}$, respectively, for MA, DMA and TMA on kaolinite at 300 K.
The $\gamma_{eff}$ of amines on kaolinite were on the same order as the $\gamma_{eff}$ of amines on coarse
particles, including $(NH_4)_2SO_4$, $NH_4NO_3$, $NH_4Cl$ and citric acid (Liu et al., 2012a;Liu
et al., 2012b) investigated in our previous work, and on 20-500 nm $NH_4NO_3$ particles
(Lloyd et al., 2009), while they were 1 and 3 orders of magnitude lower than that on the
surface of a $H_2SO_4$ solution (Wang et al., 2010a) and ammonium salt clusters (Bzdek et
al., 2010b), respectively.

There are several factors affecting the heterogeneous reaction kinetics of amines

with the typical components of atmospheric particles. First, particle size plays an
important role in the reactivity of amines. For example, the first-order $\gamma$ of DMA on 1-
2 nm $[(NH_4)_3(SO_4)_2]^+$ clusters was close to unity (Bzdek et al., 2010b), while $\gamma$
decreased to $10^{-3}$-$10^{-2}$ on coarse particles of $(NH_4)_2SO_4$ (Liu et al., 2012a;Qiu et al.,
2011). Second, strong acidity of a particle or solution favors the reactions between
amines and the substrates. The $\gamma$ of amines on citric acid, with a $pK_{a1}$ of 3.1, were 2
orders of magnitude higher than those on humic acid, with a $pK_{a1}$ of 5 (Liu et al., 2012b).
$H_2SO_4$ solution with higher $H_2SO_4$ content also showed a slightly larger $\gamma$ for amines
(Wang et al., 2010a). Even for different ammonium compounds, the H with stronger
acidity in the $NH_4$ group showed a higher reactivity toward MA (Liu et al., 2012a).
Third, the steric effect of amines has an effect on the reactivity on the same substrate.



It has been found that the γ of amines on citric acid and humic acid linearly decreased
with the cross-sectional area of the amines (Liu et al., 2012b). Finally, reaction
conditions such as temperature and humidity should have an influence on the reactivity
between amines and particles. Wang et al. (Wang et al., 2010a) observed that the γ of
amines negatively depended on temperature due to the negative temperature
dependence of the mass accommodation coefficient. This might partially explain the
difference in the measured γ of MA on $(NH_4)_2SO_4$ between our previous work (Liu et
al., 2012a) and Qiu's work (Qiu et al., 2011). Chan et al. (Chan and Chan, 2012) found
that a high RH favored the formation of TEAH sulfate in the displacement reaction
between TEA and $(NH_4)_2SO_4$. Under the same reaction conditions, the properties of
both the particles and the amines should have an effect on the heterogeneous reactivity
of amines.

Figure 4A shows a box chart of the measured $γ_{eff}$ of these three amines on kaolinite

at 300 K. Means comparisons were performed with the Dunn-Sidak method. ANOVA
analysis demonstrated a statistically significant difference in the mean $γ_{eff}$ among MA,
DMA and TMA at the 0.05 level ($F$=49.3, $P$=2.54×10$^{-13}$). The $γ_{eff}$ of MA was
significantly larger than that of DMA, while it was significantly smaller than that of
TMA. This should be ascribed to the difference in the properties of the amines, because
the influence of the particle size and acidity on the reactivity, as mentioned above, can
be ruled out when the same substrate is used. The reactivity sequence observed in this
study is in contrast with that between amines and organic acids (Liu et al., 2012b). This
might be explained by the different types of reactive sites for organic acids and kaolinite
toward amines. For organic acids, the –COOH group is the reactive site (Liu et al.,
2012b), while Lewis acid sites predominantly contribute to the uptake of amines on
kaolinite in this study, as discussed in Section 3.1. The interaction between –COOH and



amines should be more sensitive to the neighboring groups than that between $M^+$ (Lewis
acid) and amines, because the C-C bonds in organic acid are more flexible than the M-
O (metal-oxygen) bonds in kaolinite. This means that the steric effect of amines is
unimportant for the reaction between amines and kaolinite. As shown in Fig. 4B, the
$\gamma_{eff}$ of amines on kaolinite are positively correlated with the basicity of amines (H. K.
Hall, 1957). Therefore, the slightly higher reactivity of TMA on kaolinite can be
explained by its strong basicity and the lower reactivity of DMA on kaolinite should be
related to its weak basicity.
**3.3 Temperature dependence of amine uptake on kaolinite.** Table 2 summarizes the
mean $\gamma_{eff}$ at different temperature. The $\gamma_{eff}$ were in the range of $(2.27\pm0.26)\times10^{-3}$-
$(5.79\pm0.64)\times10^{-3}$  for  MA,  $(1.71\pm0.26)\times10^{-3}$-$(6.04\pm1.24)\times10^{-3}$  for  DMA  and
$(2.95\pm0.63)\times10^{-3}$-$(9.24\pm0.26)\times10^{-3}$ for TMA. For all three amines, the $\gamma_{eff}$ increased
with decreasing temperature. This is similar to the uptake of amines on the surface of
$H_2SO_4$ solution (Wang et al., 2010a), the uptake of $HNO_3$, HCl and $N_2O_5$ on the surface
of water droplets (Doren et al., 1999) and the uptake of $N_2O_5$ on the surface of
$(NH_4)_2SO_4$ (Griffiths and Anthony Cox, 2009).
Uptake of amines on kaolinite is the result of the coupled processes of mass
accommodation (gas-to-particle transfer) and reaction with the reactive sites (mainly
Lewis acid) on the surface of kaolinite. Thus,
$$\frac{1}{\gamma} = \frac{1}{\alpha} + \frac{1}{\Gamma_{rxn}} \quad (11)$$
where $\alpha$ is the mass accommodation coefficient; $\Gamma_{rxn}$ is the surface reaction resistance
(Ammann et al., 2003). If the reaction on the surface is fast enough, accommodation at
the surface of amines becomes the rate determining step (RDS) and $\gamma$ is equal to $\alpha$.
According to the thermodynamic model of mass accommodation (Davidovits et al.,
2006), the relationship between the $\alpha$ and temperature can be described as,





$\quad \ln \frac{\alpha}{1-\alpha} = \ln \frac{\gamma}{1-\gamma} = \frac{-\Delta H_{obs}}{RT} + \frac{\Delta S_{obs}}{R}$ (12)
where $R$ is the ideal gas constant; $T$ is reaction temperature. Therefore, the adsorption
enthalpy ($\Delta H_{obs}$) and entropy ($\Delta S_{obs}$) for amines on kaolinite can be obtained by
equation (12) (Liu et al., 2010b;Hudson et al., 2002;Davidovits et al., 2006). Figure 5
shows the plot of $\gamma/(1-\gamma)$ versus inverse temperature. The temperature dependence of
the $\gamma$ of amines on kaolinite is consistent with accommodation-controlled uptake with
the assumption $\gamma=\alpha$. This also satisfactorily explained the independence of the observed
uptake coefficient on sample mass as discussed in Section 3.2, because of the large $\gamma$.
$\qquad$ The $\Delta H_{obs}$ and $\Delta S_{obs}$ for adsorption of MA on kaolinite were determined to be -
$7.8\pm0.8$ kJ mol$^{-1}$ and $-77.1\pm3.2$ J K$^{-1}$ mol$^{-1}$, respectively. They were $-9.9\pm2.9$ kJ mol$^{-1}$
and $-84.1\pm11.8$ J K$^{-1}$ mol$^{-1}$ for DMA and $-9.4\pm1.0$ kJ mol$^{-1}$ and $-80.6\pm3.7$ J K$^{-1}$ mol$^{-1}$
for TMA. The $\Delta H_{obs}$ or $\Delta S_{obs}$ values are not significantly different at the 0.05 level
among the different amines. The $\Delta H_{obs}$ for uptake of amines on kaolinite are comparable
with that of carbonyl sulfide ($-10.7\pm1.1$ kJ mol$^{-1}$) on kaolinite (Liu et al., 2010b), while
they are smaller than the $\Delta H_{obs}$ of $N_2O_5$ on $(NH_4)_2SO_4$ ($-33$ KJ mol$^{-1}$) (Griffiths and
Anthony Cox, 2009) and $N_2O_5$ on sulfuric acid aerosol ($-25$ KJ mol$^{-1}$) (Hallquist et al.,
2000). The relative small $\Delta H_{obs}$ values of amines on kaolinite demonstrate that amines
weakly adsorb on kaolinite.
$\qquad$ The empirical formulas relating $\gamma_t$ of amines on kaolinite and temperature are given
as
$\quad \gamma_{eff}(\text{MA}) = \frac{\exp(938.8/T-9.3)}{1+\exp(938.8/T-9.3)}$ (13)
$\quad \gamma_{eff}(\text{DMA}) = \frac{\exp(1193.8/T-10.1)}{1+\exp(1193.8/T-9.3)}$ (14)
$\quad \gamma_{eff}(\text{TMA}) = \frac{\exp(1126.5/T-9.6)}{1+\exp(1126.5/T-9.6)}$ (15)
Thus, the $\gamma_t$ at other temperatures can be obtained using these equations.



**4 Conclusions and environmental implications**
The uptake of amines on kaolinite was investigated within a temperature range of 232-
300 K. It was found that Lewis acid sites on kaolinite were the main contributors to the
uptake of amines. The initial effective uptake coefficients of amines were
$(2.27\pm0.26)\times10^{-3}$, $(1.71\pm0.26)\times10^{-3}$ and $(2.95\pm0.63)\times10^{-3}$, respectively, for MA, DMA
and TMA on kaolinite at 300 K. The basicity of amines was weakly correlated with the
reactivity at 300 K, namely, TMA, with the strongest basicity, showed the highest
reactivity on kaolinite, and vice versa for DMA. The uptake coefficients increased ~3-
fold with decreasing temperature from 300 K to 232 K. Based on the temperature
dependence of the uptake coefficients, the uptake of amines was predominantly
ascribed to mass accommodation. The $\Delta H_{obs}$ of MA, DMA and TMA on kaolinite were
$-7.8\pm0.8$, $-9.9\pm2.9$ and $-9.4\pm1.0$ kJ mol$^{-1}$, respectively. The corresponding $\Delta S_{obs}$ were
$-77.1\pm3.2$, $-84.1\pm11.8$ and $-80.6\pm3.7$ J K$^{-1}$ mol$^{-1}$. The empirical formula relating $\gamma$ and
temperature can be expressed as shown in Eqs. (13)-(15).
With the measured uptake coefficients, the lifetimes of amines attributable to
uptake by kaolinite can be estimated by
$$\tau = \frac{4}{\gamma_{eff}\bar{v}SA} \quad (16)$$
where $\bar{v}$ is the average velocity of amines (m s$^{-1}$); $\gamma_{eff}$ is the effective uptake
coefficient at 300 K; and $SA$ is the surface area of particles in the dust event (m$^2$ m$^{-3}$).
If we assume that all mineral dust is in the form of kaolinite, and the dust loading is 150
$\mu$m$^2$ cm$^{-3}$ (de Reus et al., 2000; Frinak et al., 2004) under extreme conditions, the
lifetimes of MA, DMA and TMA due to heterogeneous uptake onto dust were estimated
to be 7.2, 11.5 and 7.7 h, respectively. In the atmosphere, oxidation by OH was
considered to be the main degradation pathway of amines. The second-order rate
constants of aliphatic amines toward OH are $(2-6)\times10^{-11}$ cm$^{-3}$ molecule$^{-1}$ s$^{-1}$ (Atkinson,





1986). Thus, their lifetimes are in the range of 4.6-13.8 h with the assumption of a 24 h
average OH concentration of $1.0\times10^6$ molecules cm$^{-3}$ (Prinn et al., 2001). Therefore,
the estimated lifetimes of amines related to heterogeneous uptake by mineral dust are
comparable to those consumed by OH oxidation. Of course, the contribution of
heterogeneous uptake to the amine sink might be overestimated here, because the
uptake coefficient decreased with exposure time quickly as shown in Fig. 1. It should
be noted that the $\gamma_t$ values of amines on kaolinite increased significantly at low
temperature as shown in Table 2. On the other hand, high concentrations of mineral
dust are possible near particular emission sources, such as agriculture, which is an
important source of amines (Kuhn et al., 2011). Therefore, mineral dust may have an
important influence on the local concentration of amines, especially in low temperature
regions with high concentrations of mineral dust. Finally, uptake experiments were
performed in a Knudsen cell reactor in this study. Although water vapor was also
introduced into the reactor along with amines, the RH was still lower than 1 %. If high
RH can also promote the uptake of amines on mineral dust as observed on $(NH_4)_2SO_4$,
$NH_4NO_3$ and $NH_4Cl$ (Chan and Chan, 2012), the contribution of mineral dust to the
amine sink will be enhanced.
Recent studies have found that alkylaminium sulfates with lower vapor pressure
than that of ammonium sulfate are more thermally stable than their counterparts (Lavi
et al., 2013) and the displacement reactions of alkylamines with ammonium sulfate lead
to a transition from the crystalline to an amorphous phase, improved water uptake (Qiu
and Zhang, 2012) and CCN activity (Lavi et al., 2013), and less scattering ability for
360 and 420 nm radiation (Lavi et al., 2013). This means that heterogeneous reactions
greatly modify the properties of aerosols (Gomez-Hernandez et al., 2016). Kaolinite
particles have been confirmed as effective ice nuclei (IN) (Wex et al., 2014;Salam et al.,



2006), while physically adsorbed amines on kaolinite can reduce the water wettability
of kaolinite (Kitahara and Williamson, 1964). This means that heterogeneous uptake of
amines on kaolinite may have an influence on the IN ability of kaolinite, although this
needs to be confirmed with experiments in the future.
Alkylaminium sulfate salts, even in freshly nucleated nanoparticles (lower than 10
nm in diameter), will not be likely to undergo particle to gas partitioning at room
temperature because of their thermal stability and ultralow vapor pressures (Lavi et al.,
2013). However, physical adsorption with small enthalpy takes place between amines
and kaolinite. This means that adsorbed amines probably at or near the emission sources
of amines can be transported with dust to other regions, and subsequently might
undergo particle-to-gas partitioning in air with a lower concentration of amines, and
subsequently, further participate in new particle formation with acids or displacement
reactions with ammonium compounds. In this process, mineral dust takes on the role of
a carrier or reservoir of amines.

**Acknowledgements**
This research was financially supported by the National Natural Science Foundation of
China (41275131) and the Strategic Priority Research Program of Chinese Academy of
Sciences (XDB05040100, XDB05010300).

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





**Tables**

Table 1. Comparison of the $\gamma$ of amines on typical particles.

| Amines | Particles | $\gamma$ | T (K) | Ref. |
|---|---|---|---|---|
| MA | Kaolinite | $2.27\pm0.26\times10^{-3}$ | 300 | This study |
| | Citric acid [a] | $7.31\pm1.13\times10^{-3}$ | 300 | (Liu et al., 2012b) |
| | Humic acid [a] | $1.26\pm0.07\times10^{-5}$ | 300 | (Liu et al., 2012b) |
| | $(NH_4)_2SO_4$ [a] | $6.30\pm1.03\times10^{-3}$ | 300 | (Liu et al., 2012a) |
| | $(NH_4)_2SO_4$ [b] | $2.60\text{-}3.40\times10^{-2}$ | 293 | (Qiu et al., 2011) |
| | $NH_4HSO_4$ [a] | $1.78\pm0.36\times10^{-2}$ | 300 | (Liu et al., 2012a) |
| | $NH_4NO_3$ [a] | $8.79\pm1.99\times10^{-3}$ | 300 | (Liu et al., 2012a) |
| | $NH_4Cl$ [a] | $2.29\pm0.28\times10^{-3}$ | 300 | (Liu et al., 2012a) |
| | 62% $H_2SO_4$ solution | $2.00\pm0.20\times10^{-2}$ | 283 | (Wang et al., 2010a) |
| | 71% $H_2SO_4$ solution | $2.00\pm0.30\times10^{-2}$ | 283 | (Wang et al., 2010a) |
| | 82% $H_2SO_4$ solution | $3.00\pm0.50\times10^{-2}$ | 283 | (Wang et al., 2010a) |
| DMA | Kaolinite | $1.71\pm0.26\times10^{-3}$ | 300 | This study |
| | Citric acid [a] | $6.65\pm0.49\times10^{-3}$ | 300 | (Liu et al., 2012b) |
| | Humic acid [a] | $7.33\pm0.40\times10^{-6}$ | 300 | (Liu et al., 2012b) |
| | $(NH_4)_2SO_4$ [b] | $3.20\text{-}3.40\times10^{-2}$ | 293 | (Qiu et al., 2011) |
| | $[(NH_4)_3(SO_4)_2]^+$ cluster [c] | $0.85\pm0.26$ | r.t. | (Bzdek et al., 2010b) |
| | $[(NH_4)_2(HSO_4)]^+$ cluster [c] | $1.05\pm0.26$ | r.t. | (Bzdek et al., 2010b) |
| | $[(NH_4)_3(HSO_4)_2]^+$ cluster [c] | $0.85\pm0.22$ | r.t. | (Bzdek et al., 2010b) |
| | $[(NH_4)_4(HSO_4)_3]^+$ cluster [c] | $0.61\pm0.15$ | r.t. | (Bzdek et al., 2010b) |
| | $[(NH_4)_3(NO_3)_2]^+$ cluster [c] | $0.53\pm0.21$ | r.t. | (Bzdek et al., 2010b) |
| | 30% $(NH_4)_2SO_4$ solution | $1.80\text{-}0.60\times10^{-2}$ | 293 | (Qiu et al., 2011) |
| | 62% $H_2SO_4$ solution | $3.00\pm0.60\times10^{-2}$ | 283 | (Wang et al., 2010a) |
| | 71% $H_2SO_4$ solution | $2.50\pm0.40\times10^{-2}$ | 283 | (Wang et al., 2010a) |
| | 82% $H_2SO_4$ solution | $3.20\pm0.30\times10^{-2}$ | 283 | (Wang et al., 2010a) |
| TMA | Kaolinite | $2.95\pm0.63\times10^{-3}$ | 300 | This study |
| | Citric acid [a] | $5.82\pm0.68\times10^{-3}$ | 300 | (Liu et al., 2012b) |
| | Humic acid [a] | $4.75\pm0.15\times10^{-6}$ | 300 | (Liu et al., 2012b) |
| | $(NH_4)_2SO_4$ [b] | $2.40\text{-}2.90\times10^{-2}$ | 293 | (Qiu et al., 2011) |
| | $[(NH_4)_2(HSO_4)]^+$ cluster [c] | $0.90\pm0.26$ | r.t. | (Bzdek et al., 2010b) |
| | $[(NH_4)_3(HSO_4)_2]^+$ cluster [c] | $0.66\pm0.26$ | r.t. | (Bzdek et al., 2010b) |
| | $[(NH_4)_4(HSO_4)_3]^+$ cluster [c] | $0.64\pm0.26$ | r.t. | (Bzdek et al., 2010b) |
| | $[(NH_4)_3(NO_3)_2]^+$ cluster [c] | $0.40\pm0.12$ | r.t. | (Bzdek et al., 2010b) |
| | $NH_4NO_3$ [d] | $2.00\pm2.00\times10^{-3}$ | r.t. | (Lloyd et al., 2009) |
| | 62% $H_2SO_4$ solution | $2.20\pm0.20\times10^{-2}$ | 283 | (Wang et al., 2010a) |
| | 71% $H_2SO_4$ solution | $2.70\pm0.80\times10^{-2}$ | 283 | (Wang et al., 2010a) |
| | 82% $H_2SO_4$ solution | $3.50\pm0.20\times10^{-2}$ | 283 | (Wang et al., 2010a) |

Note: Note: [a] coarse particles (grounded crystalline); [b] coarse particles (crystalline or
amorphous from solution); [c] 1-2 nm particles; [d] 20-500 nm particles





Table 2. The $\gamma_{eff}$ of amines on kaolinite in the temperature range 232-300 K

| Temperature (K) | $\gamma_{eff}$ (MA) | $\gamma_{eff}$ (DMA) | $\gamma_{eff}$ (TMA) |
|---|---|---|---|
| 232 | $5.79 \pm 0.64 \times 10^{-3}$ | $6.04 \pm 1.24 \times 10^{-3}$ | $9.24 \pm 0.26 \times 10^{-3}$ |
| 237 | $4.70 \pm 0.58 \times 10^{-3}$ | - | - |
| 248 | $3.90 \pm 0.86 \times 10^{-3}$ | $5.35 \pm 1.03 \times 10^{-3}$ | $5.90 \pm 0.93 \times 10^{-3}$ |
| 258 | - | - | $4.40 \pm 0.03 \times 10^{-3}$ |
| 263 | $3.51 \pm 0.39 \times 10^{-3}$ | $4.68 \pm 1.23 \times 10^{-3}$ | - |
| 278 | $2.63 \pm 0.30 \times 10^{-3}$ | $2.25 \pm 0.46 \times 10^{-3}$ | $3.44 \pm 0.40 \times 10^{-3}$ |
| 300 | $2.27 \pm 0.26 \times 10^{-3}$ | $1.71 \pm 0.26 \times 10^{-3}$ | $2.95 \pm 0.63 \times 10^{-3}$ |






**Figure captions**
**Figure 1.** Uptake curves and the corresponding uptake coefficient of amines on
kaolinite at 300 K. The sample mass was 19.3, 20.6 and 20.2 mg, respectively.
**Figure 2.** *In situ* FTIR spectra of amines adsorbed on kaolinite for 30 min at 300 K.
The IR spectra were recorded taking the dried kaolinite as reference.
**Figure 3.** Mass dependence of $\gamma_{obs}$ for amines on kaolinite at 300 K. Error bars indicate
1 σ for repeated experiments.
**Figure 4.** (A) Box chart for the $\gamma_{eff}$ of amines on kaolinite measured at 300 K. (B)
Relationship between the $\gamma_{eff}$ and the basicity of amines.
**Figure 5.** Temperature dependence of $\gamma_{eff}$ for amines on kaolinite.






**Figures**

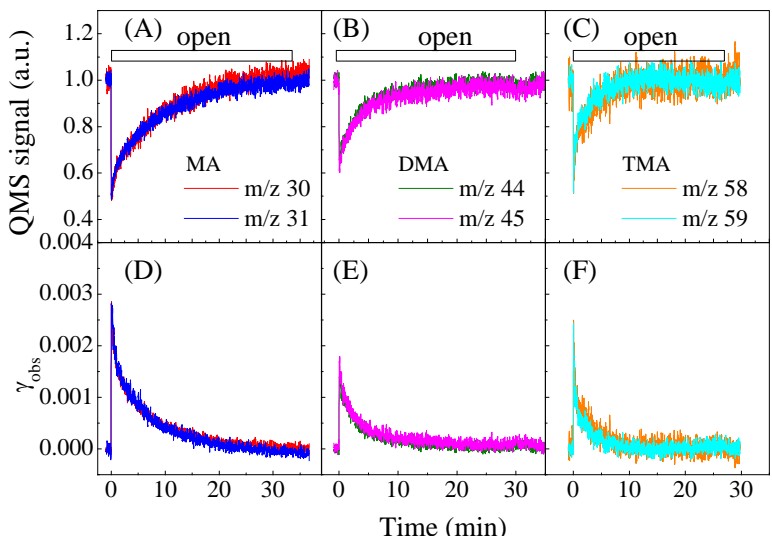

**Fig. 1**



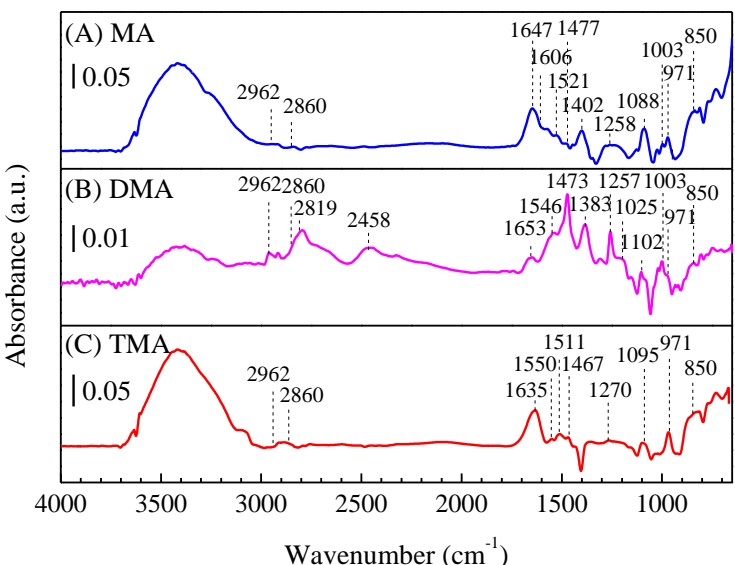


Wavenumber (cm⁻¹)

**Fig. 2**





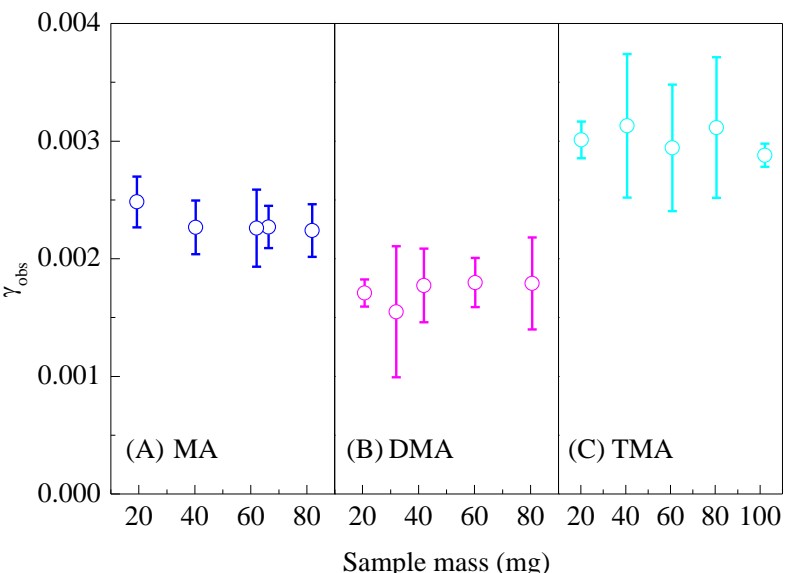


**Fig. 3**





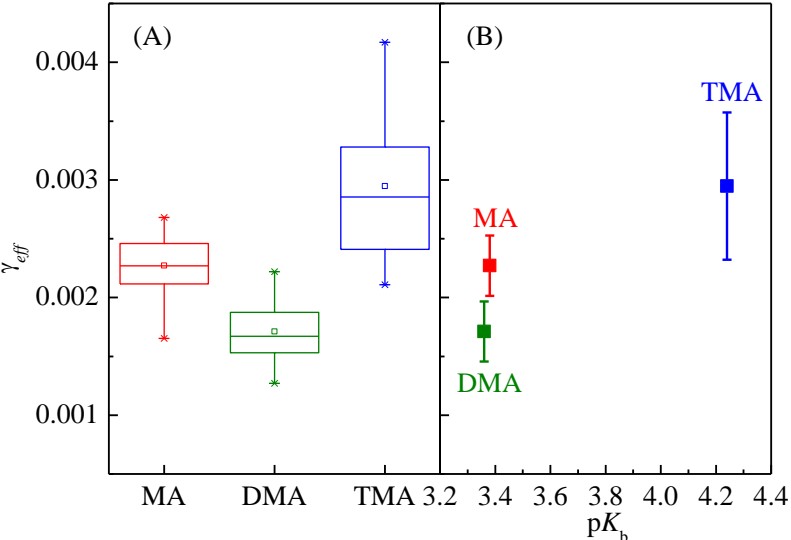


**Fig. 4**





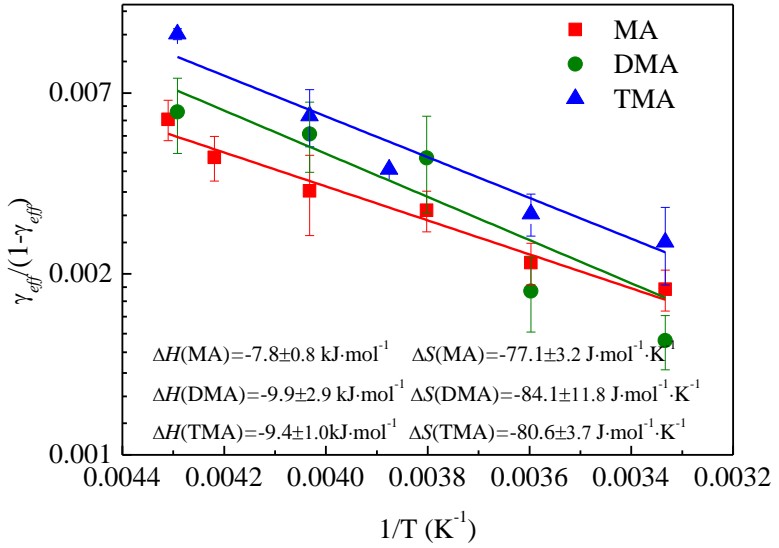


**Fig. 5**