# Peer review of "Heterogeneous uptake of amines onto kaolinite in the"

_Atmospheric Chemistry and Physics, 2016_

## Referee Comment (RC1) · Anonymous Referee #1 · 26 Jul 2016

The authors report uptake coefficients of three amines (methylamine MA, dimethylamine DMA, trimethylamine TMA) on a mineral dust model substance (clay mineral kaolinite) using a Knudsen reactor in the temperature range 232-300 K. Using a simple thermodynamic model for non-reactive systems (P. Davidovits) they go on to interpret the negative T-dependence in terms of adsorption enthalpies and entropies. The paper reads well, however, the authors do not direct their attention to conveying a take-home lesson from their experiments because there are too many loose ends that are not explained or interpreted correctly. Almost all experiments leave wanting to such an extent that the level and quality of effort is not commensurate with the expectations of the readers of acp. The problem is that both kinetics and spectroscopy parts are "light", thus incomplete and misleading. Some of the conclusions are not supported by experimental fact and are left hanging. An example is the assertion that the amines

interact with Lewis acid sites on the kaolinite despite the fact that the authors insist on identifying protonated amines in the DRIFTS spectra which would correspond to Bronsted acidic sites on kaolinite, unless I misunderstand. However, only DMA shows protonated base (Figure 2), no evidence is presented for TMA and MA. This is just an example for the severe flaws and shortcomings of this report that prevents the reader from understanding the initial question posed and its eventual resolution. I conclude that his paper is nowhere near to being publishable and I therefore list the most important questions and concerns for the benefit of the authors that need to be answered to the satisfaction of the editor before publication may be envisaged.

I would like to begin with several technical questions that need to be answered such that the interested reader may see for himself rather than taking the author's word for it: trust is good, control is better!

- I have not found a diagram of the Knudsen reactor in any of the author's papers. I assume, that their machine does not include molecular beam modulation and recovery of the chopped signal using a lock-in amplifier. This is usually a fatal flaw in case one is dealing with "sticky" compounds like the present amines. I strongly suggest that the authors include a block diagram (as an Appendix) on the design of the instrument.

- The authors do not seem to ever perform (or report the results of) reference (blank) experiments with the empty sample compartment exposed to the amines. Owing to the day-long (8 h) saturation of the internal vessel walls with the amines one cannot be sure about the interpretation of the signal lasting just 10 to 20 minutes during which "saturation" of the substrate takes place. This is definitely the wrong instrument to tackle the problem at hand.

- I am missing a Table with all salient parameters of the Knudsen reactor such as gas-wall collision frequency, volume and surface of the flow reactor, sample surface, used escape rate constant, flow rate calibrations, MS sensitivities, etc.

- Regarding the low-temperature runs I am missing a detailed sketch (again in the Appendix) of the cooling module: are the feeding lines for the coolant insulated inside the Knudsen reactor? Are the authors sure that the amines are not exposed to additional cold surfaces in low temperature runs? Again, let the reader see ALL the DETAILS! Let the reader make the decision as to the validity of the chosen experimental set-up. Most importantly, I have not seen any blanks with the empty cold cell at the lowest temperatures used. This is a must in order to instill a minimum of confidence in your experiments. In addition, the authors withhold any experimental uptake curves at low temperatures which would be of significant interest to many colleagues!

- Have the authors checked whether or not the uptake corresponds to a first-order rate-law? If the rate law is more complex, and I suspect it is, how good an approximation is a first-order rate law? The Knudsen reactor is a suitable instrument to check out the rate law: one has to vary the rate constant of escape by varying the orifice diameter: if the rate constant for uptake is independent of the orifice diameter, thus the gas residence time, then we have a first-order rate law. Have the authors varied the flow rate? By the way, what was the flow rate of the amine into the flow reactor? One may see that these questions cannot remain unanswered for a halfway complete and reasonable experimental kinetic study on the uptake of amines on kaolinite.

- First and foremost I am missing a calibration of the residual gas MS signals in terms of amine concentrations. To that end I do not understand why the authors use an aqueous solution of the amines. In order to calibrate the amine signals they must use pure amines which are commercially available. One cannot interpret saturation curves such as Figure 1 if one does not have the slightest idea how many molecules of amine it takes to saturate the kaolinite: Does the saturation level correspond to a fraction of a monolayer, one or several layers? These are important questions in order to interpret and grasp the meaning of these saturation experiments.

The following questions are more general:

- Pg. 9, top: What is the typical lifetime of MA, DMA or TMA on the stainless steel

vessel walls? How does the signal vary with time if you interrupt the flow of amine at once? This should provide the characteristic residence time of the amines on the vessel walls, at least at the beginning because the MS signal decays are complex, unimolecular at the beginning, and more complex at the end.

- Pg. 10, lines 224-226: What is the experimental evidence for the interaction of the amines with Lewis acidic sites on the kaolinite. The same message is recurrent: See also pg. 15, line 359, pg. 16, line 379 and pg. 18, line 412. What is the experimental proof or physical evidence for this? Protonated amines originate from the neutralization of Bronsted, not Lewis acid sites! Furthermore, it is unjustified to postulate protonated MA and TMA from Figure 2. There are no peaks at 1467-1477 cm-1 for these two amines that I can see! To that effect, the authors must amplify and expand the spectrum such that one may distinguish the noise from a potential absorption FTIR signal in the above spectral range.

- How did the authors evaluate the numbers on pg. 11, line 250 (molecules mg-1) in the complete absence of any quantitative calibration of the amine MS signals?

- Regarding Figure 3: The plateau of gamma(obs) seems to be reached for a mass of kaolinite ranging from 20 to 100 mg. Why does the situation change that much for COS/kaolinite (ref. 2010b) in Figure 4 (or reference 2010b) where a sample mass of 20 mg is definitely at the beginning of the linear mass regime (LMR)? In contrast to V. Grassian and coworkers the LMR may be interpreted also in the sense that the mass is not sufficient to cover up the sample surface area with a coherent material layer because there are "holes" in the substrate layer. This depends of course on the particle size. What was the particle size in this study? This information should go into the technical Table requested above.

- On pg. 13 the authors evaluate gamma(eff) using the KML theory. What are the values of the parameters dp, "tau", $\eta$, $\varphi$, etc. so as to be able to follow the authors in their calculation.

- Pg. 14, line 324 and following: What is the reason the particle size plays such a large role for the magnitude of the uptake coefficient? This question comes up several times without the authors giving an answer.

- It does not make sense to correlate ïA̧ğgammaeff with pKb of the amines in Figure 4. The latter parameter is dominated by solvent effects because DMA is a stronger base than TMA in solution whereas one expects the inverse. What the authors should take is either the proton affinity (PA or enthalpy of protonation in the gas phase) or gas phase basicity (gB or equilibrium constant). The values are: NH3 (853.6/819.0 kJ/mol corresponding to PA/gB), MA (899.0/864.5), DMA (929.5/896.5), TMA (948.9/918.1). In this series TMA is clearly the strongest base which is an intrinsic property of the molecule compared to MA and DMA.

- Pg. 17, enthalpy and entropy of vaporization: The resulting thermodynamic parameters do not make any sense at all as they are at least a factor of three too small compared to the experimental heat of vaporization of the amines: MA (25.6 kJ/mol), DMA (26.4), TMA (22.94). If the values of the present study were true, then why should the amines interact with kaolinite at all? They certainly will prefer to condense unto itself onto the stainless steel walls into small droplets rather than to adsorb on kaolinite! The reason is that equation (2) is too simple a model for this reactive system. Rather, one must distinguish physisorption from chemisorption. Davidovits did not develop his simple model to a reactive system, therefore, it seems that the simple model is totally inadequate and yields unphysical results.

- Pg. 17, line 404: What did you fit in order to obtain equations (13) to (15)?

- Pg. 19, middle: What is the saturation behavior of the amines at low temperature?

- Pg. 18, line 419-420: "…the uptake of amines was predominantly ascribed to mass accommodation" is hard to understand because mass accommodation is seldom rate-limiting, but transition over a barrier is.

- Table 2, pg. 26: Show raw data at low temperatures!

- Pg. 32, Figure 5: The TMA data lie on a curve, NOT on a straight line! There are important deviations

Some of less important items:

- Pg. 3, line 41: What are "anthropological" emissions?

- Pg. 6, line 138: Tabor et al., 1994, Ullerstam et al., 2003: references are missing.

- Pg. 13, 295: Salgado-Muñoz and„„also in bibliographic list at the end (line 623)!

- Pg. 14, line 327: cluster has the wrong polarity!

- Pg. 14, line 332: Larger than what? The author's comparison only has one leg!!

- Pg. 15, line 344: What is TEAH sulfate?

- Pg. 20, line 473: mineral dust is a bad reservoir or no reservoir at all! The authors should cut out qualitative or meaningless talk.

- Pg. 23, line 653: "Physiochemical"?

- Pg. 25, Table 1: Under DMA: fifth entry from the top of DMA field has wrong polarity!

Please also note the supplement to this comment:
http://www.atmos-chem-phys-discuss.net/acp-2016-538/acp-2016-538-RC1-supplement.pdf

———————————————

---

## Referee Comment (RC2) · Anonymous Referee #2 · 4 Aug 2016

Using a Knudesen cell reactor and ATR-FTIR, Liu et al. investigated the heterogeneous reactions of methylamine (MA), dimethylamine (DMA), and trimethylamine (TMA) with kaolinite (as a surrogate of mineral dust) and the effect of temperature. Both amines and mineral dust are important components in the troposphere, and their reations have not been examined yet. This manuscript fits the scope of ACP well and the results are quite new. The kinetic data present by this work would help us better understand the tropospheric sinks of amines and the aging processes of mineral dust particles. This manuscript can be published after the following comments are addressed:

**Major comments:**

Line 133-136: Prior to the uptake measurement, the reacton chamber was passivated with amines to reduce/minimize the wall effect. The sample chamber also has some (though smaller compared to the reaction chamber) wall effect. Is this significant compared to the uptake by kaolinite? I believe this can be determined by backgroud experiments in which no dust is depoisted onto the sample holder.

Line 429-438: While I agree with the authors that heterogeneous reactions with mineral dust can be an important sink for these amines in the troposphere, I also two comments: 1) the effect of gas phase diffusion needs to be discussed (Tang et al., 2015), especially for large particles (e.g., mineral dust) and fast uptake (which is also the case in this study); 2) only extreme conditions with very high dust loadings are discussed; to understand the general role of these reactions, the authors should also discuss the lifetimes under typical atmospheric conditions. By the way, the dust loading unit used in this manuscript is $\mu m2\ cm-3$; while this is convenient to calculate the lifetime using Eq. (16), the corresponding mass concentration (which is more widely used) should also be provided.

Line 467-468: It is stated that physical adsorption takes place between amines and kaolinites, but no direct experimental evidence is provided. As I understand, both Knudsen cell reactor and ATR-FTIR can be use to examine whethere a gas-surface reaction is reversile. I would suggest that another 1-2 figures with experimental data should be included to clarify this issue.

**Minor comments:**

Line 4: I should suggest that "amines" should be changed to "methylamine, dimethylamine, and trimethylamine (TMA)" to be specific.

Line 24-25: This statement is incorrect. The uptake coefficients were directly derived from the experimental data as discussed in Sections 3.1 and 3.2, and mass accommodation coeffeicients are used to derive enthalpies and entropies (Section 3.3).

Line 49: please also provide the concentrations in pptv.

Line 81: The review paper by Crowley et al. (2010) should also be cited here together with Usher et al. (2003).

Line 312: I believe "$\gamma_{eff}$" should be "$\gamma_{eff}/\gamma_{obs}$".

Line 339-343: It should be further explained why the study by Wang et al. (2010a) explained the difference between Liu et al. (2012a) and Qiu et al. (2011). For the current manuscript, it is not clear to me.

Line 457-463: the effects of heterogeneous reactions on the chemical composition and IN activity of mineral dust particles is mentioned here. I do believe that it should also be mentioned in the introduction. Besides, many papers have discussed the effects of heterogeneous reaction on the hygroscopicity and CCN and IN activities of mineral dust, including those by Cziczo et al. (2009), Sullivan et al. (2009), Ma et al. (2012), Tobo et al. (2012) and Tang et al. (2016), just to name a few.

Figure 5: It will improve the readability of this figure to move $\Delta H$ and $\Delta S$ values to the figure caption instead.

**Reference**

Crowley, J. N., Ammann, M., Cox, R. A., Hynes, R. G., Jenkin, M. E., Mellouki, A., Rossi, M. J., Troe, J., and Wallington, T. J.: Evaluated Kinetic and Photochemical Data for Atmospheric Chemistry: Volume V - Heterogeneous Reactions on Solid Substrates, Atmos. Chem. Phys., 10, 9059-9223, 2010.

Cziczo, D. J., Froyd, K. D., Gallavardin, S. J., Moehler, O., Benz, S., Saathoff, H., and Murphy, D. M.: Deactivation of ice nuclei due to atmospherically relevant surface coatings, Environ. Res. Lett., 4, 044013, 2009.

Ma, Q. X., Liu, Y. C., Liu, C., and He, H.: Heterogeneous Reaction of Acetic Acid on MgO, α-Al2O3, and CaCO3 and the Effect on the Hygroscopic Behavior of These Particles, Phys. Chem. Chem. Phys., 14, 8403-8409, 2012.

Sullivan, R. C., Moore, M. J. K., Petters, M. D., Kreidenweis, S. M., Roberts, G. C., and Prather, K. A.: Timescale for Hygroscopic Conversion of Calcite Mineral Particles through Heterogeneous Reaction with Nitric Acid, Phys. Chem. Chem. Phys., 11, 7826-7837, 2009.

Tang, M. J., Shiraiwa, M., Pöschl, U., Cox, R. A., and Kalberer, M.: Compilation and evaluation of gas phase diffusion coefficients of reactive trace gases in the atmosphere: Volume 2. Diffusivities of organic compounds, pressure-normalised mean free paths,

and average Knudsen numbers for gas uptake calculations, Atmos. Chem. Phys., 15, 5585-5598, 2015.

Tang, M. J., Cziczo, D. J., and Grassian, V. H.: Interactions of Water with Mineral Dust Aerosol: Water Adsorption, Hygroscopicity, Cloud Condensation and Ice Nucleation, Chem. Rev., 116, 4205–4259, 2016.

Tobo, Y., DeMott, P. J., Raddatz, M., Niedermeier, D., Hartmann, S., Kreidenweis, S. M., Stratmann, F., and Wex, H.: Impacts of chemical reactivity on ice nucleation of kaolinite particles: A case study of levoglucosan and sulfuric acid, Geophys. Res. Lett., 39, L19803, doi: 19810.11029/12012gl053007, 2012.

Wagner, C., Hanisch, F., Holmes, N., de Coninck, H., Schuster, G., and Crowley, J. N.: The interaction of N2O5 with mineral dust: aerosol flow tube and Knudsen reactor studies, Atmos. Chem. Phys., 8, 91-109, 2008.

---

## Author Comment (AC1) · 22 Sep 2016

**Referee #1**

The authors report uptake coefficients of three amines (methylamine MA, dimethylamine DMA, trimethylamine TMA) on a mineral dust model substance (clay mineral kaolinite) using a Knudsen reactor in the temperature range 232-300 K. Using a simple thermodynamic model for non-reactive systems (P. Davidovits) they go on to interpret the negative T-dependence in terms of adsorption enthalpies and entropies. The paper reads well, however, the authors do not direct their attention to conveying a take-home lesson from their experiments because there are too many loose ends that are not explained or interpreted correctly. Almost all experiments leave wanting to such an extent that the level and quality of effort is not commensurate with the expectations of the readers of acp. The problem is that both kinetics and spectroscopy parts are "light", thus incomplete and misleading. Some of the conclusions are not supported by experimental fact and are left hanging. An example is the assertion that the amines interact with Lewis acid sites on the kaolinite despite the fact that the authors insist on identifying protonated amines in the DRIFTS spectra which would correspond to Bronsted acidic sites on kaolinite, unless I misunderstand. However, only DMA shows protonated base (Figure 2), no evidence is presented for TMA and MA. This is just an example for the severe flaws and shortcomings of this report that prevents the reader from understanding the initial question posed and its eventual resolution. I conclude that his paper is nowhere near to being publishable and I therefore list the most important questions and concerns for the benefit of the authors that need to be answered to the satisfaction of the editor before publication may be envisaged.

**Response**: Thank you for your comments and suggestions. We reply to your particular concerns about the IR results here. Other questions will be answered point-by-point below.

We stated that "In this work, two bands at 1477-1467 cm$^{-1}$ and 971 cm$^{-1}$ were observed, as shown in Fig. 2, and were probably related to the protonated amines (MAH$^+$, DMAH$^+$ and TMAH$^+$). In particular, a strong peak at 1473 cm$^{-1}$ was observed (Fig. 2B), while the band was very weak for the spectra in Fig. 2A and C. However, this band was very close to the δ(NH) band at ~1470 cm$^{-1}$ in DMA (NIST database) and/or δ(CH$_3$) (Lin et al., 2014) in amines. On the other hand, it should be noted that formation of protonated amines requires surface OH (Brønsted acid). However, as shown in Fig. 2, the surface hydroxyl (-OH) in the range of 3600-3750 cm$^{-1}$ (Miranda-Trevino and Coles, 2003) was not consumed when the kaolinite was exposed to MA, DMA or TMA. This indicates that the content of reactive Brønsted acid sites on the kaolinite must be very low even if the bands at 1477-1467 cm$^{-1}$ originated from the protonated amines. Therefore, Lewis acid sites on the kaolinite predominantly contributed to the uptake of amines as observed in Figs. 1 and 2 in this study. This was similar to the adsorption of DMA and N,N-dimethyl formamide (DMF) on TiO$_2$, for which Lewis acid sites were identified as the reactive sites (Lin et al., 2014)." in lines 210-227.

In Fig. 2, although two IR bands (1473 and 971 cm$^{-1}$) were close to the characterized bands of protonated TMA ((CH$_3$)$_3$NH$^+$) at 1484 and 984 cm$^{-1}$, the band at 1484 cm$^{-1}$ was also close to the δ(NH) band of Lewis-bound DMA and/or the δ(CH$_3$) band at ~1470 cm$^{-1}$ in DMA. In addition, the consumption of OH, which was required to form protonated DMA, was not observed during adsorption of DMA. Therefore, the contribution of protonated amines to the band at 1473 cm$^{-1}$ was ruled out.

To make our statements more clear, we will revise our manuscript about the IR results as "For MA in Fig. 2A, the band at 1606 cm$^{-1}$ (-NH$_2$ deformation band) is the characteristic band for Lewis-bound MA (Auerbach et al., 2003;Nunes et al., 2005). In Fig. 2B, the band at ~1470 cm$^{-1}$ might have originated from the δ(NH) in Lewis-bound DMA (NIST database) and/or δ(CH$_3$) (Lin et al., 2014). In a previous work, the IR

bands at 1484 and 984 cm$^{-1}$ were assigned to the characteristic bands of protonated TMA ($(CH_3)_3NH^+$) in an acidic solution containing TMA and on the surface of polyethylene (PE) treated by TMA (Ongwandee et al., 2007). In this work, although two bands at 1477-1467 cm$^{-1}$ and 971 cm$^{-1}$ were observed in the spectrum shown in Fig. 2B, and were close to the bands of protonated TMA (TMAH$^+$), it was more reasonable to assign them to Lewis-bound DMA for the following reasons. Formation of protonated amines requires the partition of surface OH (Brønsted acid). However, as shown in Fig. 2B, the surface hydroxyl (-OH) in the range of 3600-3750 cm$^{-1}$ (Miranda-Trevino and Coles, 2003) was not consumed when the kaolinite was exposed to DMA. In particular, the bands at 1477-1467 and 971 cm$^{-1}$ were not observed for MA and TMA in Fig. 2A and C. At the same time, OH consumption was also unobservable for MA and TMA. Therefore, it can be concluded that Lewis acid sites on the kaolinite predominantly contributed to the uptake of MA, DMA and TMA as observed in Figs. 1 and 2 in this study. This was similar to the adsorption of DMA and N,N-dimethyl formamide (DMF) on TiO$_2$, for which Lewis acid sites were identified as the reactive sites (Lin et al., 2014)".

I would like to begin with several technical questions that need to be answered such that the interested reader may see for himself rather than taking the author's word for it: trust is good, control is better! I have not found a diagram of the Knudsen reactor in any of the author's papers. I assume, that their machine does not include molecular beam modulation and recovery of the chopped signal using a lock-in amplifier. This is usually a fatal flaw in case one is dealing with "sticky" compounds like the present amines. I strongly suggest that the authors include a block diagram (as an Appendix) on the design of the instrument.

**Response**: Thank you so much for your good suggestions. Yes, our system does not include molecular beam modulation and a lock-in amplifier. The block diagram (Fig. R1) will be included in Fig. S1. In the experimental details, this will be pointed out as "The schematic diagram is shown in Fig. S1. The parameters of the reactor are summarized in Tables S1.".

[Figure]

**Fig. R1**. The schematic diagram of the Knudsen cell-mass spectrometer.

As you know, there are two types of Knudsen cell reactors. 1)With molecular beam modulation and a lock-in amplifier (Seisel et al., 2005;Seisel et al., 2004;Ullerstam et al., 2003;Salgado-Muñoz and Rossi, 2002;Fenter et al., 1994); 2)Without molecular beam modulation and lock-in amplifier (Hoffman et al., 2003;Underwood et al., 2001;Beichert and Finlayson-Pitts, 1996;Liu et al., 2008).

Molecular beam modulation can distinguish between signals in the mass spectrometer arising from background gases in the vacuum system and those produced by the desorbed species by modulating either the adsorbed or desorbed molecular beam and measuring the resulting time-dependent signal in the mass spectrometer. Under this circumstance, it requires a detector capable of recovering signals in the presence of an overwhelming noise background or, alternatively, providing high resolution measurements of relatively clean signals over several orders of magnitude and frequency. Thus, a lock-in amplifier is usually combined with molecular beam modulation. By combination with these techniques, a Knudsen cell reactor with molecular beam modulation and a lock-in amplifier can directly measure the surface lifetime of an adsorbed gas, the order of chemical reaction, and the accommodation coefficient and sticking coefficient of incident molecules (Compton et al., 1984). The uptake experiments using such an instrument can be operated in both a steady flow

mode and a pulse mode. By contrast, only the net uptake coefficient can be measured, usually in a steady flow, with a Knudsen cell reactor without molecular beam modulation and a lock-in amplifier (Underwood et al., 2000). This means that a Knudsen cell reactor with molecular beam modulation and a lock-in amplifier should be more powerful than the counterpart without these features. However, the net uptake coefficient measured in this work and those reported in the literature with the second type of Knudsen cell are also meaningful in atmospheric chemistry.

The uptake coefficient ($\gamma_{net}$) is defined as,

$$\gamma \overset{def}{=} \frac{-\dfrac{dn_g}{dt}}{\omega} \quad (R1)$$

where $n_g$ is number of molecules in the gas phase (molecules), $t$ is time (s), and $\omega$ is gas-surface collision frequency ($s^{-1}$). When the pressure in the reactor cell is kept low enough, the mean free path of the molecules within the cell is greater than the cell dimensions, so that boundary layer effects and homogeneous gas-phase collisions are minimized and can be neglected. Gas-surface collision frequency is given by

$$\omega = \frac{\bar{v}A_s}{4}\frac{n}{V} = \frac{vA_s}{4}\frac{4F}{A_h v} = \frac{FA_s}{A_h} \quad (R2)$$

where $\bar{v}$ is the average molecular speed ($m\ s^{-1}$); $A_s$ is the area that the flux of molecules impinges upon or the geometric area of the sample holder ($m^2$); ($n/V$) is the number density of the reactant gas (molecules $m^{-3}$); $F$ is the flow of molecules out of the cell (molecules $s^{-1}$); $A_h$ is the area of the exit aperture ($m^2$).

In a steady-state flow, the number of the reactant molecules that is "lost" to the surface ($-dn_g/dt$) is equal to the change in flow out of the cell, ($F_0 - F$), where $F_0$ and $F$ represent the gas-phase flow out of the cell with the sample covered and exposed, respectively (Underwood et al., 2000). Therefore,

$$\gamma = \frac{A_h}{A_s}\frac{F_0 - F}{F} \quad (R3)$$

Once the gas-phase flow out of the cell is proportional to the measured mass spectral intensity of the reactant gas ($I$),

$$\gamma = \frac{A_h}{A_s} \frac{I_0 - I}{I} \quad \text{(R4)}$$

Therefore, Eq. (R4) is correct and without any additional limitations if the uptake experiment is performed in a molecular flow (or the mean free path of the molecules within the cell is greater than the cell dimensions) and in a steady-state flow, and the gas flow out of the cell is proportional to the measured mass spectral intensity of the reactant gas ($I$).

In this work, the working pressure was ~$3.5 \times 10^{-4}$ Torr. The mean free path of gas was greater than 12 cm even at 232 K. This value was greater than the maximal dimensions of the cell (5.6 cm). By adjusting the variable leak valve, the mass signal was calibrated against the pressure in the cell, which was proportional to the flow rate (Table R1). As shown in Fig. R2, the signal intensity of TMA linearly responded to the flow rate or the pressure in the reactor. Similar results were also obtained for MA and DMA.

**Table R1**. Calibration of the MS signal of TMA to the flow rate.

| No. | Pressure in the reactor (Torr) | Signal intensity (m/z 59) | σ(m/z 59) | Signal intensity (m/z 58) | σ(m/z 58) |
|---|---|---|---|---|---|
| P1 | 1.00E-03 | 5.90E+04 | 1.16E+03 | 2.91E+04 | 7.76E+02 |
| P2 | 1.30E-04 | 8.27E+03 | 4.38E+02 | 4.02E+03 | 3.02E+02 |
| P3 | 2.50E-04 | 1.72E+04 | 5.97E+02 | 8.43E+03 | 4.28E+02 |
| P4 | 3.70E-04 | 2.38E+04 | 7.16E+02 | 1.17E+04 | 5.20E+02 |
| P5 | 7.40E-04 | 3.79E+04 | 1.09E+03 | 1.86E+04 | 6.83E+02 |
| P6 | 3.90E-04 | 2.35E+04 | 7.10E+02 | 1.15E+04 | 5.02E+02 |
| P7 | 3.10E-04 | 1.80E+04 | 6.19E+02 | 8.83E+03 | 4.53E+02 |
| P8 | 1.80E-04 | 1.16E+04 | 5.45E+02 | 5.66E+03 | 3.73E+02 |
| P9 | 1.00E-04 | 6.30E+03 | 3.89E+02 | 3.02E+03 | 2.70E+02 |
| P10 | 6.00E-05 | 3.40E+03 | 3.03E+02 | 1.61E+03 | 1.96E+02 |
| P11 | 2.40E-05 | 9.21E+02 | 1.59E+02 | 3.73E+02 | 1.21E+02 |

[Figure]

**Fig. R2**. The relationship between the MS signal and the pressure in the reactor or the flow rate of TMA. "*I*" and "*F*" means the MS signal intensity and the flow rate of TMA.

We agree with you that amines are sticky molecules. We compared the escape rate of amines with that of $N_2$ (in the same experiment). The escape rate ($k_{Esc}$) was calculated by

$$\ln \frac{I - I'}{I_0 - I'} = -k_{Esc}t \quad (R5)$$

Where $I_0$ and $I'$ is the signal intensity at the initial and the final steady state when the flow rate into the reactor was changed using the variable leak valve. As shown in Fig. R3, the measured escape rate of TMA was slightly smaller than that of $N_2$ at 300 K. However, if the effective area of the escape hole ($A_h$) was calculated based on amines but not $N_2$, the effect of the viscosity of amines on the measurement of uptake coefficient should be ruled out according to Eq. (R4) because the kinematic viscosity has been factored into the $k_{Esc}$ or $A_h$ calculation. The $A_h$ was obtained by

$$A_h = \frac{4V}{\tau \bar{v}} = \frac{4V k_{Esc}}{\bar{v}} \quad (R6)$$

 summarizes the measured effective $A_h$ of these three amines after each uptake experiment was completed. The effective $A_h$ of amines were significantly smaller than that of $N_2$. In particular, the mean $A_h$ of MA was the smallest one among these three amines. This is in agreement with the larger coefficient of kinematic viscosity of MA (13.9 $m^2$ $s^{-1}$) compared with that of DMA (8.7 $m^2$ $s^{-1}$) and TMA (6.7 $m^2$ $s^{-1}$) calculated at 300 K and $3.5 \times 10^{-4}$ Torr based on the Peng-Robinson model (Fan and Wang, 2006).

[Figure]

**Fig. R3**. Escape rate of TMA and $N_2$ measured at 300 K.

**Table R2**. The area of escape hole ($A_h$) of amines measured at 300 K. Unit: $mm^2$.

| Exp. No. | MA | $N_2$ | DMA | $N_2$ | TMA | $N_2$ |
|---|---|---|---|---|---|---|
| 1 | 0.28 | 0.89 | 0.29 | 0.80 | 0.76 | 0.91 |
| 2 | 0.32 | 1.12 | 0.42 | 0.89 | 0.61 | 0.93 |
| 3 | 0.30 | 1.15 | 0.41 | 0.97 | 0.81 | 1.08 |
| 4 | 0.28 | 0.94 | 0.44 | 0.87 | 0.75 | 0.83 |
| 5 | 0.25 | 0.77 | 0.38 | 0.78 | 0.62 | 0.82 |
| Average | 0.28±0.03 | 0.97±0.16 | 0.39±0.06 | 0.86±0.07 | 0.71±0.09 | 0.92±0.10 |

In the original manuscript, we calculated the uptake coefficients based on the $A_h$ of $N_2$. This would overestimate the uptake coefficients of amines. In the revised manuscript, all the uptake coefficients have been corrected based on the measured $A_h$ of the amines at the corresponding temperature and pressure. The corrections will also be done in our previous work. In the experimental section, it will be discussed in the **section 3.2** as "It should be pointed out that the escape rate of amines from the reactor should be lower than that of $N_2$ because amines are sticky molecules. This will lead to a smaller effective $A_h$ of amines compared with $N_2$. Table S2 summarizes the summarized $A_h$ for these amines at 300 K and $3.5 \times 10^{-4}$ Torr. The effective $A_h$ of amines were significantly smaller than that of $N_2$. At the same time, MA showed the smallest $A_h$ among these three amines. This is in agreement with the larger coefficient of kinematic viscosity of MA (13.9 $m^2$ $s^{-1}$) compared with that of DMA (8.7 $m^2$ $s^{-1}$) and TMA (6.7 $m^2$ $s^{-1}$) calculated at 300 K and $3.5 \times 10^{-4}$ Torr based on the Peng-Robinson model (Fan and Wang, 2006). Therefore, the effective $A_h$ of amines but not $N_2$ measured at the corresponding temperature and pressure were used to calculate the uptake coefficients according to Eq. (1). The $A_h$ at different temperatures are summarized in Tables S2-S3".

The authors do not seem to ever perform (or report the results of) reference (blank) experiments with the empty sample compartment exposed to the amines. Owing to the day-long (8 h) saturation of the internal vessel walls with the amines one cannot be sure about the interpretation of the signal lasting just 10 to 20 minutes during which "saturation" of the substrate takes place. This is definitely the wrong instrument to tackle the problem at hand.

**Response**: We have performed blank experiments for all of these amines. Fig. R4 shows the results (raw data) for the blank experiments. This experiment was carried out at 232 K. When the steady state of MA or TMA signals was reached, the Teflon sample holder (without particles) was exposed to MA or TMA vapor by opening the sample cover. As shown in Fig. R4, the signals of MA or TMA did not change. This means the contribution of the fresh surface (including the sample holder and the inner surface of

the sample cover) to the uptake of amines by kaolinite is negligible. The blank experiment results will be added in the supporting information as Fig. S2. In the manuscript, it will also be pointed out that "Blank experiments showed that the fresh surface (including the sample holder and the inner surface of the sample cover) did not contribute to the signal drops observed for amines when the sample cover was opened (Fig. S2)."

[Figure]

**Fig. R4**. Blank experiments (raw data) for (A) MA and (B) TMA at 232 K.

In the experimental section, we mentioned that the sample was out-gassed at 298 K in the Knudsen cell reactor for 8 h (lines 127-130, in the original manuscript) but not passivated with amines for 8 h. We also pointed out that the reactor chamber was passivated with amines while the sample was isolated from the reactant gas by the sample cover, until a steady state QMS signal was established (lines 133-135, in the original manuscript). The passivation time was usually 1 h. In the revised manuscript, this will be described as "The reactor chamber was passivated with amines while the sample was isolated from the reactant gas by the sample cover, until a steady state QMS signal was established (~ 60 min)".

I am missing a Table with all salient parameters of the Knudsen reactor such as gas wall collision frequency, volume and surface of the flow reactor, sample surface, used escape rate constant, flow rate calibrations, MS sensitivities, etc.

**Response**: Thank you for your suggestion. The volume ($V$), surface of the inner wall ($A_w$), the diameter ($\Phi$), the height ($H$) and the sample surface ($A_s$) of the reactor, the area of the escape hole ($A_h$), the escape rate constant ($k_{esc}$), the flow rate of amine ($F$) and the MS sensitivity have been summarized in Table R3. These parameters of the Knudsen cell reactor will be included in the supporting information Table S1. As discussed above, the $A_h$ of amines varied among these three amines and temperature, while it was constant within the uncertainty for $N_2$ in the range of 232-300 K. Therefore, the $A_h$ of $N_2$ was listed in this table. The $A_h$ of amines will be summarized in Tables S2 and S3. In the manuscript, this will be pointed out as "The parameters of the reactor are summarized in Tables S1".

**Table R3**. The parameters of the Knudsen cell reactor.

| Parameter | value |
|---|---|
| $V$ (cm$^3$) | 93.90 |
| $A_w$ (cm$^2$) | 116.20 |
| $\Phi$(cm) | 5.60 |
| $H$(cm) | 3.50 |
| $A_s$ (cm$^2$) | 3.26 |
| $A_h$ ($N_2$, mm$^2$) | 0.90±0.15 |
| $k_{esc}$ ($N_2$, s$^{-1}$) | 1.14±0.14 |
| $F$ of amines (molecules s$^{-1}$) | ~1.2×10$^{13}$ |
| MS sensitivity of amines (3σ, molecules s$^{-1}$) | ~1.1×10$^{12}$ (MA), ~1.4×10$^{12}$ (DMA), ~1.7×10$^{12}$ (TMA) |
| $D$p of kaolinite (μm) | 0.56 |

Although the surface of the wall was larger than that of the sample holder, the wall was inactive for amine uptake after being passivated in the steady flow as shown in Fig. R4.

Regarding the low-temperature runs I am missing a detailed sketch (again in the Appendix) of the cooling module: are the feeding lines for the coolant insulated inside the Knudsen reactor? Are the authors sure that the amines are not exposed to additional cold surfaces in low temperature runs? Again, let the reader see ALL the DETAILS! Let the reader make the decision as to the validity of the chosen experimental set-up. Most importantly, I have not seen any blanks with the empty cold cell at the lowest temperatures used. This is a must in order to instill a minimum of confidence in your experiments. In addition, the authors withhold any experimental uptake curves at low temperatures which would be of significant interest to many colleagues!

**Response**: Thank you for your suggestion. The detailed sketch of the cooling module was included in Fig. R1. There was a small isolated chamber cooled with low-temperature ethanol. Temperature was measured with a Pt resistance thermometer embedded in the top ceiling of the cooling chamber on which the sample holder sat. When the sample was exposed to amines, there was no additional cold surface except for the inner surface of the sample cover and the sample holder. The results of the blank experiments have been shown in Fig. R4. However, as shown in Fig. R4 in the blank experiment performed at 232 K, the fresh surface of the sample cover and the sample holder did not contribute to the signal changes in the presence of amines. The uptake curve of amines at 232 K is shown in Fig. R5. This figure has been added in the SI as Fig. S6. In the text, a paragraph will be added as "Fig. S6 shows the uptake curves of amines on kaolinite at 232 K. The signals of all three amines in Fig. S6 decreased more than those observed at 300 K (Fig. 1) after the samples were exposed to amines. For example, the $I/I_0$ of MA, DMA and TMA were 0.32, 0.30 and 0.22, respectively, compared with the values of 0.45, 0.63, and 0.47 at 300 K" in **section 3.3.**

[Figure]

**Fig. R5**. The uptake curves of (A) MA on 21.0 mg kaolinite, (B) DMA on 20.7 mg kaolinite and (C) TMA on 20.5 mg kaolinite at 232 K.

Have the authors checked whether or not the uptake corresponds to a first-order rate law? If the rate law is more complex, and I suspect it is, how good an approximation is a first-order rate law? The Knudsen reactor is a suitable instrument to check out the rate law: one has to vary the rate constant of escape by varying the orifice diameter: if the rate constant for uptake is independent of the orifice diameter, thus the gas residence time, then we have a first-order rate law. Have the authors varied the flow rate? By the way, what was the flow rate of the amine into the flow reactor? One may see that these questions cannot remain unanswered for a halfway complete and reasonable experimental kinetic study on the uptake of amines on kaolinite.

**Response**: Thank you for your suggestion. We have measured the uptake coefficient of TMA on kaolinite with different escape rates. As shown in Fig. R6, the uptake coefficient of TMA was independent of the escape rate under our experimental conditions, which indicates a first-order rate law.

[Figure]

**Fig. R6** The dependence of uptake coefficient on escape rate of TMA. T: 300 K, mass of kaolinite: 20 mg.

It should be pointed out that the flow rate into the reactor was too small (~$10^{-9}$ L min$^{-1}$) to directly measure because we do not have a sufficiently sensitive apparatus to do the flow rate calibration. The flow rate could be measured based on the pressure differential across the constriction produced by an orifice plate if the size of the orifice is known. However, the accurate orifice size was unavailable because a variable leak valve with non-circular hole was used in this work. Therefore, we estimated the flow rate based on the law of conservation of mass. According to the pumping rates of the turbomolecular pumps, the corresponding pressure and the vapor pressure of amines over the solution, the flow rate of amines was estimated to be ~$1.2 \times 10^{13}$ molecules s$^{-1}$. The MS sensitivity was estimated to be ~$1.1 \times 10^{12}$, ~$1.4 \times 10^{12}$ and ~$1.7 \times 10^{12}$ molecules s$^{-1}$ for MA, DMA and TMA, respectively, based on the standard deviation of the MS signal ($3\sigma$). In the experimental section, a sentence will be added as "Based on the law of conservation of mass, the flow rate of amines into the reactor was around $1.2 \times 10^{13}$ molecules s$^{-1}$ according to the pumping rates of the turbomolecular pumps, the

corresponding pressure and the saturation vapor pressure of the amines" in section 2.1.

First and foremost I am missing a calibration of the residual gas MS signals in terms of amine concentrations. To that end I do not understand why the authors use an aqueous solution of the amines. In order to calibrate the amine signals they must use pure amines which are commercially available. One cannot interpreted saturation curves such as Figure 1 if one does not have the slightest idea how many molecules of amine it takes to saturate the kaolinite: Does the saturation level correspond to a fraction of a monolayer, one or several layers? These are important questions in order to interpret and grasp the meaning of these saturation experiments.

**Response**: e agree with you that pure amines should be used to calibrate the MS signals. Actually, a permeate tube is required to perform calibration for amines and ammonia (Freshour et al., 2014;Neuman et al., 2003). However, the pure amine cylinders are unavailable in China, and a permeate tube for amines is also unavailable in our laboratory at the present time. Thus, we used the amine solutions to generate amine vapors.

It should be noted that the relative concentration of amine is required to measure the uptake coefficient according to Eq. (R3) or Eq. (R4). Once the MS signal linearly responds to the flow rate or concentration of amine, Eq. (R4) is correct. As shown in Fig. R2, the MS signal was linearly correlated to the flow rate of amine into the reactor or the pressure in the reactor. This means that the measured uptake coefficients of amines in this work are credible.

Of course, the absolute flow rate or concentration of amine is required to calculate the uptake capacity of amine on kaolinite. As stated above, the flow rate of the amines into the reactor was estimated to be $\sim 1.2 \times 10^{13}$ molecules $s^{-1}$ based on the law of conservation of mass with the pumping rates, the corresponding pressures and the saturation vapor pressure of the amines. Therefore, the uptake capacity can be roughly estimated. We carefully checked the calculation method for the uptake capacity. An improper integration method was utilized in our original manuscript and a conversion factor was missed. The correct uptake capacities of MA, DMA and TMA on kaolinite

were ~$1.4\times10^{14}$, ~$6.3\times10^{13}$ and ~$3.7\times10^{13}$ molecules mg$^{-1}$, respectively. This has been corrected in the revised manuscript. In addition, the total amines lost on the kaolinite surface (in Fig. 1) were ~$2.7\times10^{15}$, ~$1.3\times10^{15}$ and ~$7.4\times10^{14}$ molecules, respectively, under our experimental conditions. With the assumption of the cross-sectional area of MA (0.243 nm$^2$), DMA (0.323 nm$^2$), and TMA (0.394 nm$^2$) (Liu et al., 2012), the adsorbed amines would cover 6.6 (MA), 4.2 (DMA) and 2.9 cm$^2$ (TMA) in a monolayer. These values were comparable to the geometric area of the sample holder (3.26 cm$^2$) and less than the surface area of the kaolinite (1.4 m$^2$). Therefore, the saturation level shown in Fig. 1 might correspond to a fraction of a monolayer under our conditions. In the revised manuscript, we will add the related discussion as "The integrated uptake capacities were obtained based on the uptake curves shown in Fig. 1 once the MS signal intensities were converted to the flow rate of amines into the reactor. The uptake capacities were ~$2.7\times10^{15}$, ~$1.3\times10^{15}$ and ~$7.4\times10^{14}$ molecules for MA, DMA and TMA, respectively, under our experimental conditions. The areas of the adsorbed amines in a monolayer would be 6.6 (MA), 4.2 (DMA) and 2.9 cm$^2$ (TMA). These values are comparable to the geometric area of the sample holder (3.26 cm$^2$) and less than the surface area of the kaolinite (1.42 m$^2$). This means that the saturation level in Fig. 1 might correspond to a fraction of a monolayer under our conditions".

The following questions are more general:

Pg. 9, top: What is the typical lifetime of MA, DMA or TMA on the stainless steel vessel walls? How does the signal vary with time if you interrupt the flow of amine at once? This should provide the characteristic residence time of the amines on the vessel walls, at least at the beginning because the MS signal decays are complex, unimolecular at the beginning, and more complex at the end.

**Response**: The residence time of N$_2$ ($\tau=1/k_{esc}$) was 0.90±0.15 s$^{-1}$ in the temperature range of 323-300 K, while the values were slightly dependent on temperature for MA, DMA and TMA (Fig. R7). The mean residence time ($\tau$), the escape rate ($k_{esc}$) and the area of the escape hole ($A_h$) for amines and N$_2$ at different temperatures based on the data fitting are summarized in Table R4. This table has been added in the supporting

information as .

[Figure]

**Fig. R7**. Temperature dependence of the residence time in the temperature range of 232-300 K.

**Table R4**. The mean residence time ($\tau$), the escape rate ($k_{esc}$) and the area of the escape hole ($A_h$) in the temperature range of 232-300 K.

| T(K) | $\tau$(s) | | | | $k_{esc}(s^{-1})$ | | | | $A_h(mm^2)$ | | | |
|------|------|------|------|------|------|------|------|------|------|------|------|------|
| | MA | DMA | TMA | $N_2$ | MA | DMA | TMA | $N_2$ | MA | DMA | TMA | $N_2$ |
| 232 | 2.22 | 1.73 | 1.27 | | 0.46 | 0.56 | 0.79 | | 0.43 | 0.63 | 1.02 | |
| 237 | 2.28 | 1.84 | 1.29 | | 0.45 | 0.54 | 0.78 | | 0.42 | 0.61 | 1.00 | |
| 248 | 2.41 | 2.07 | 1.32 | | 0.43 | 0.50 | 0.76 | | 0.39 | 0.56 | 0.97 | |
| 258 | 2.53 | 2.29 | 1.36 | 1.14 | 0.41 | 0.47 | 0.75 | 0.90 | 0.37 | 0.51 | 0.93 | 0.93 |
| 263 | 2.58 | 2.39 | 1.37 | | 0.40 | 0.45 | 0.74 | | 0.36 | 0.49 | 0.92 | |
| 278 | 2.76 | 2.71 | 1.42 | | 0.38 | 0.40 | 0.72 | | 0.33 | 0.42 | 0.87 | |
| 300 | 3.02 | 3.18 | 1.49 | | 0.34 | 0.32 | 0.69 | | 0.28 | 0.32 | 0.79 | |

Pg. 10, lines 224-226: What is the experimental evidence for the interaction of the amines with Lewis acidic sites on the kaolinite. The same message is recurrent: See also pg. 15, line 359, pg. 16, line 379 and pg. 18, line 412. What is the experimental

proof or physical evidence for this? Protonated amines originate from the neutralization of Bronsted, not Lewis acid sites! Furthermore, it is unjustified to postulate protonated MA and TMA from Figure 2. There are no peaks at 1467-1477 cm-1 for these two amines that I can see! To that effect, the authors must amplify and expand the spectrum such that one may distinguish the noise from a potential absorption FTIR signal in the above spectral range.

**Response**: As a primary amine, Lewis-bound MA gives rise to a characteristic $\delta$ band at 1606 cm$^{-1}$ (-NH$_2$ deformation band) (Auerbach et al., 2003;Nunes et al., 2005). In Fig. 2A, the band at 1606 cm$^{-1}$ was observable for the uptake of MA on kaolinite. Lewis-bound DMA shows the $\delta$(NH) band at ~1470 cm$^{-1}$ (Lin et al., 2014), which was observed in Fig. 2B. Lewis-bound TMA does not have the typical $\delta$ band of NH$_x$ except for the IR bands related to CH$_3$ and NC$_3$. For all of these amines, however, Lewis-bound amines can be indirectly confirmed by the absence of OH consumption during uptake. As you noted, no peaks at 1467-1477 cm$^{-1}$ were observable for MA and TMA (Fig. 2A and C). This further confirmed that they were adsorbed on the Lewis acid sites. Physisorption has been confirmed by repeat uptake experiments (Fig. R10) and will be discussed below.

We will revise the description of the IR spectra as "For MA in Fig. 2A, the band at 1606 cm$^{-1}$ (-NH$_2$ deformation band) is the characteristic band for Lewis-bound MA (Auerbach et al., 2003;Nunes et al., 2005). In Fig. 2B, the band at ~1470 cm$^{-1}$ might have originated from the $\delta$(NH) in Lewis-bound DMA (NIST database) and/or $\delta$(CH$_3$) (Lin et al., 2014). In a previous work, the IR bands at 1484 and 984 cm$^{-1}$ were assigned to the characteristic bands of protonated TMA ((CH$_3$)$_3$NH$^+$) in an acidic solution containing TMA and on the surface of polyethylene (PE) treated by TMA (Ongwandee et al., 2007). In this work, although two bands at 1477-1467 cm$^{-1}$ and 971 cm$^{-1}$ were observed in the spectrum shown in Fig. 2B, and were close to the bands of protonated DMA (DMAH$^+$), it was more reasonable to assign them to Lewis-bound DMA for the following reasons. Formation of protonated amines requires the partition of surface OH (Brønsted acid). However, as shown in Fig. 2B, the surface hydroxyl (-OH) in the range of 3600-3750 cm$^{-1}$ (Miranda-Trevino and Coles, 2003) was not consumed when the

kaolinite was exposed to DMA. For MA and TMA, the bands at 1477-1467 and 971 cm$^{-1}$ are discernable in Fig. 2A and C. At the same time, OH consumption was also unobservable when kaolinite was exposed to MA or TMA. Therefore, it can be concluded that Lewis acid sites on the kaolinite predominantly contributed to the uptake of MA, DMA and TMA as observed in Figs. 1 and 2 in this study. This is similar to the adsorption of DMA and N,N-dimethyl formamide (DMF) on $TiO_2$, for which Lewis acid sites were identified as the reactive sites (Lin et al., 2014)".

How did the authors evaluate the numbers on pg. 11, line 250 (molecules mg-1) in the complete absence of any quantitative calibration of the amine MS signals?

**Response**: T These numbers were estimated using the flow rate of amines based on the law of conservation of mass in the light of the pumping rates of the turbomolecular pumps, the corresponding pressure and the vapor pressure of the amines over the solution. As pointed out above, standard gas sources of amines were unavailable. The flow rate was roughly estimated here. Using a similar method, we have estimated the flow rate into the reactor of $SO_2$, COS and amines in our previous work (Ma et al., 2012;Liu et al., 2012;Liu et al., 2008). Although the uncertainties in the flow rate from the pumping rate, pressure and vapor pressure still remain, the trend of the estimated adsorption capacity among these three amines in this work should be correct. The method to estimate the flow rate of amines into the reactor will be added in the experimental section as "Based on the law of conservation of mass, the flow rate of amines into the reactor was around $1.2 \times 10^{13}$ molecules s$^{-1}$ according to the pumping rates of the turbomolecular pumps, the corresponding pressure and the saturation vapor pressure of the amines".

Regarding Figure 3: The plateau of gamma(obs) seems to be reached for a mass of kaolinite ranging from 20 to 100 mg. Why does the situation change that much for COS/kaolinite (ref. 2010b) in Figure 4 (or reference 2010b) where a sample mass of 20 mg is definitely at the beginning of the linear mass regime (LMR)? In contrast to V. Grassian and coworkers the LMR may be interpreted also in the sense that the mass is

not sufficient to cover up the sample surface area with a coherent material layer because there are "holes" in the substrate layer. This depends of course on the particle size. What was the particle size in this study? This information should go into the technical Table requested above.

**Response**: Thank you. This difference can be explained by the different diffusion depths of the reactant gas in the multilayer particles. As discussed in our manuscript (**section 3.2**), the probe depth of a reactant in a packed powder sample is governed by the bulk density, the true density, the tortuosity factor and the particle diameter of the sample. It is also determined by the effective uptake coefficient ($\gamma_{eff}$) of the reactant. According to Eq. (8) and (9), η (a factor to account for the effect of gas-phase diffusion into the underlying layers) can be simplified as

$$\eta = C \cdot \gamma_{eff}^{-1/2} \cdot \tanh(\gamma_{eff}^{1/2}) \quad \text{(R7)}$$

where *C* is a constant and a function of the bulk density, the true density, the tortuosity factor and the particle diameter of the sample, the sample mass and the geometric area of the sample holder. A large $\gamma_{eff}$, which is determined by the properties of both the reactant and the substrate (particles), corresponds to a small $\eta$. An $\eta$ with zero value means that the underlying layers of the packed powder sample do not contribute to the uptake of the reactant gas.

The minimal *m* was 20 mg; $\rho_b$ was measured as 0.65 g cm$^{-3}$; $\rho_t$ was 2.62 g cm$^{-3}$ (Wardhana et al., 2014); $d_p$ was 0.56 μm; $A_g$ was 3.26 cm$^2$; τ was 3 (Matthews et al., 1996). Therefore, the relationship between $\eta$ and $\gamma_{eff}$ can be calculated with these parameters. Fig. R8 shows the relationship between $\eta$ and $\gamma_{eff}$. If the $\gamma_{eff}$ of amines on kaolinite was ~10$^{-3}$ as discussed in this paper, the $\eta$ value should be ~0.1. Strictly speaking, a nonzero $\eta$ indicates that underlying layers can contribute to the uptake of reactant gas. However, this cannot be measured if the slope of the observed uptake coefficient against the sample mass is too small, as observed in this study. Therefore, we conclude that the contribution of underlying layers of kaolinite to the uptake of amines was negligible. If the parameters of mineral oxides in our previous work were similar to those of kaolinite in this study, the $\gamma_{eff}$ of COS of ~10$^{-6}$-10$^{-7}$ (Liu et al., 2010a)

should correspond to an $\eta$ of ~1. Therefore, different responses of $\gamma_{obs}$ against sample mass can be explained by the different values of $\eta$ between the two reaction systems. In the manuscript, a paragraph will be added as "According to the minimal sample mass ($m$= 20 mg), the measured $\rho_b$ (0.65 g cm$^{-3}$), the $\rho_t$ (2.62 g cm$^{-3}$) (Wardhana et al., 2014) and the measured $d_p$ (0.56 µm) of kaolinite, the $A_g$ (3.26 cm$^2$) and the $\tau$ (3) of kaolinite reported in the literature (Matthews et al., 1996), the relationship between $\eta$ and $\gamma_{eff}$ was obtained based on Eq. (8)-(10) and is shown in Fig. S5. If the $\gamma_{eff}$ of amines are close to their $\gamma_{obs}$ reported here, the corresponding $\eta$ values should be ~0.1. On the other hand, based on the saturated adsorption capacity of amines estimated above, only 1~2 monolayers of kaolinite particles contribute to the uptake of amines. This is consistent with the assumption that uptake of amines dominantly takes place on the first layer of the kaolinite sample and the contribution of the underlying layers to amine uptake is negligible. Therefore, it is reasonable to conclude that the $\gamma_{eff}$ of amines on kaolinite are close to or equal to the $\gamma_{obs}$ in this study".

[Figure]

**Fig. R8**. The relationship between $\eta$ and $\gamma_{eff}$ on kaolinite.

Fig. R9 shows the particle size of kaolinite used in this work. The mean diameter was 0.56 μm with a half-width of 0.47 μm. This value will be added in Table S1. The TEM results will also be added in Fig. S3. A sentence will be added in the revised manuscript as "The mean diameter of kaolinite particles was 0.56 μm with a half-width of 0.47 μm (Fig. S3) measured with a transmission electron microscope (TEM, Hitachi H7500 with an accelerating voltage of 80 kV)."

[Figure]

[Figure]

**Fig. R9**. (A) TEM image of kaolinite; (B) size distribution of kaolinite particles.

With the measured bulk density (0.65 g cm$^{-3}$), the depth of the sample was 0.09 mm in the sample holder for 20 mg of kaolinite. Kaolinite particle samples with multilayers should be prepared. We tried to prepare a sample with lower mass. However, it was difficult to obtain an even film covering all of the sample holder surface. We agree with you that LMR may be interpreted in the sense that the mass is not sufficient to cover up the sample surface area with a coherent material layer because there are "holes" in the substrate layer. This was the reason why we did not perform uptake experiments with sample masses lower than 20 mg. In our work, we are sure that all of the sample holder was covered by the samples.

On pg. 13 the authors evaluate gamma(eff) using the KML theory. What are the values of the parameters dp, "tau", _, ', etc. so as to be able to follow the authors in their calculation.

**Response**: In Eq. (9), the minimum $m$ was 20 mg; $\rho_b$ was measured as 0.65 g cm$^{-3}$; $\rho_t$ was 2.62 g cm$^{-3}$ (Wardhana et al., 2014); $d_p$ was 0.56 μm; $A_g$ was 3.26 cm$^2$; τ was 3

(Matthews et al., 1996). We have added these parameters and the corresponding discussion to the revised manuscript as mentioned above.

Pg. 14, line 324 and following: What is the reason the particle size plays such a large role for the magnitude of the uptake coefficient? This question comes up several times without the authors giving an answer.

**Response**: According to the definition of uptake coefficient (Eq. R1), $\gamma$ is proportional to the net loss rate of amine. Surface reactions take place between amines and ammonium salts. Therefore, ammonium salts with smaller particle size should lead to a higher loss rate of amines due to the larger specific area when compared with the counterpart. The above discussion will be added in the revised manuscript as "According to the definition of uptake coefficient, $\gamma$ is proportional to the net loss rate of amine, and consequently, the specific area of the particles. Therefore, ammonium salts with smaller particle size should lead to a higher loss rate of amines due to the larger specific area when compared with the counterpart".

It does not make sense to correlate $\gamma_{eff}$ with pKb of the amines in Figure 4. The latter parameter is dominated by solvent effects because DMA is a stronger base than TMA in solution whereas one expects the inverse. What the authors should take is either the proton affinity (PA or enthalpy of protonation in the gas phase) or gas phase basicity (gB or equilibrium constant). The values are: NH3 (853.6/819.0 kJ/mol corresponding to PA/gB), MA (899.0/864.5), DMA (929.5/896.5), TMA (948.9/918.1). In this series TMA is clearly the strongest base which is an intrinsic property of the molecule compared to MA and DMA.

**Response**: Thank you for your good suggestion. The proton affinities (PA) are 879, 924 and 942 kJ mol$^{-1}$ for MA, DMA and TMA, respectively (Tunon et al., 1992). These values are close to those you mentioned. Therefore, the relationship between the reactivity of amines and the basicity of gaseous amines (Fig. R10B) has been discussed as you suggested. The corresponding discussion will be revised in the manuscript as "On the other hand, the basicity of amines might be another factor affecting the

reactivity. The proton affinity (PA), which is a measurement of the basicity in the gas phase, is 879, 924 and 942 kJ mol⁻¹ for MA, DMA and TMA (Tunon et al., 1992), respectively. The sequence of the reactivity is consistent with the PA of amines in the temperature range of 232-300 K. Therefore, the higher reactivity of TMA on kaolinite can be explained by its strong basicity, and vice versa for MA."

[Figure]

**Fig. R10** (A) Box chart for the $\gamma_{eff}$ of amines on kaolinite measured at 300 K. (B) Relationship between the $\gamma_{eff}$ and the proton affinity of amines.

Pg. 17, enthalpy and entropy of vaporization: The resulting thermodynamic parameters do not make any sense at all as they are at least a factor of three too small compared to the experimental heat of vaporization of the amines: MA (25.6 kJ/mol), DMA (26.4), TMA (22.94). If the values of the present study were true, then why should the amines interact with kaolinite at all? They certainly will prefer to condense unto itself onto the stainless steel walls into small droplets rather than to adsorb on kaolinite! The reason is that equation (2) is too simple a model for this reactive system. Rather, one must distinguish physisorption from chemisorption. Davidovits did not develop his simple model to a reactive system, therefore, it seems that the simple model is totally

inadequate and yields unphysical results.

**Response**: The $\Delta H_{obs}$ of MA, DMA and TMA on kaolinite were -11.2 $\pm$ 0.6, -17.2 $\pm$ 3.7 and -12.1 $\pm$ 1.3 kJ mol$^{-1}$, respectively. These values are slightly lower than the heat of vaporization ($\Delta H_{vap}$) of amines. The $\Delta H_{vap}$ represents the evaporation heat of amines from liquid phase to gas phase, while the $\Delta H_{obs}$ is the adsorption heat from the gas phase to surface-adsorbed amines. These are two different processes. As discussed above and in the manuscript, a fraction of a monolayer of amines formed on the kaolinite surface. Therefore, $\Delta H_{obs}$ reflects the strength for the interaction between the amines and kaolinite, but not the intermolecular interaction among amine molecules. On the other hand, the fresh sample holder did not lead to a decrease of MS signals, as shown in Fig. R4 in the blank experiment. All experiments were performed in a steady flow after the reactor wall had been passivated by amines. The reactor walls should also not contribute to the signal changes when the sample was exposed to amines. Therefore, the enthalpy and entropy of adsorption are meaningful. It is also useful to obtain an empirical equation of $\gamma_{eff}$ with temperature.

It should be pointed out that the enthalpy of physisorption is characterized by change in enthalpy less than 40 kJ mol$^{-1}$, while the enthalpy of chemisorption is greater than 40 kJ mol$^{-1}$ (De Moor et al., 2008). Therefore, the uptake of amines on kaolinite can be ascribed to physisorption but not chemisorption. On the other hand, we performed repeat uptake experiments after the kaolinite with adsorbed amines was evacuated overnight. The uptake curve in the second run coincided well with that in the first run (Fig. R11 and Fig. S4). This means that uptake of amines on kaolinite is reversible. Therefore, equation (2) is valid to describe the uptake of amines on kaolinite. In the revised manuscript, the following paragraph "Fig. S4 shows the repeat uptake curves of MA after the sample that had adsorbed amines in the first run was evacuated overnight and exposed to amines again in the second run. The uptake curve in the second run coincided well with that in the first run. This means amines are reversibly adsorbed on kaolinite" will be added before Eq. (2). Before Eq. (13), we will also revise the sentence as "The relatively small $\Delta H_{obs}$ values of amines on kaolinite demonstrate that amines physically adsorb on kaolinite, because the characteristic $\Delta H$ of

physisorption is less than 40 kJ mol$^{-1}$ (De Moor et al., 2008). This is in agreement with the reversible nature of adsorption, as shown in Fig. S4."

[Figure]

**Fig. R11**. Repeat uptake curves of MA after the used sample was evacuated overnight at 300 K. The sample mass was 19.8 mg.

Pg. 17, line 404: What did you fit in order to obtain equations (13) to (15)?

**Response**: These equations were obtained from fitting the uptake coefficients against temperature using Eq. (12). This will be pointed out as "Based on Eq. (12), the empirical formulas relating $\gamma_t$ of amines on kaolinite and temperature are given as".

Pg. 19, middle: What is the saturation behavior of the amines at low temperature?

**Response**: The saturation time was extended, as shown in Fig. R5, when compared with that in Fig. 1. This figure has been also added in the SI as Fig. S6.

Pg. 18, line 419-420: "…the uptake of amines was predominantly ascribed to mass

accommodation" is hard to understand because mass accommodation is seldom rate limiting, but transition over a barrier is.

**Response**: In general, gas-particle interaction includes the following elemental steps in sequence. 1) Mass transfer (diffusion) of the reactant(s) from the bulk fluid to the external surface of the particle; 2) Diffusion of the reactant from the pore mouth through the particle pores to the immediate vicinity of the internal surface; 3) Adsorption of reactant A onto the particle surface; 4) Reaction on the surface; 5) Desorption of the products from the surface; 6) Diffusion of the products from the interior of the particle to the pore mouth at the external surface; 7) Mass transfer of the products from the external particle surface to the bulk fluid. The mass accommodation includes the first two steps. As discussed in the manuscript, only physisorption of amines occurs on the surface of kaolinite. It is a barrierless step. According to Eq. (11), mass accommodation is possible if the surface interaction is fast enough. Actually, this has been observed in many reaction systems including uptake of carbonyl sulfide on kaolinite (Liu et al., 2010b), $N_2O_5$ on $(NH_4)_2SO_4$ (Griffiths and Anthony Cox, 2009), and $N_2O_5$ on sulfuric acid aerosol (Hallquist et al., 2000).

Table 2, pg. 26: Show raw data at low temperatures!

**Response**: This will be added in the Fig. S6 as mentioned above.

Pg. 32, Figure 5: The TMA data lie on a curve, NOT on a straight line! There are important deviations.

**Response**: Thank you. The data in Fig. 5 were linearly fitted. The correlation coefficients were 0.99 (MA), 0.94 (DMA), and 0.98 (TMA). We think it is reasonable to depict as a line but not a curve. The correlation coefficients will be added in the figure caption in the revised manuscript as "The correlation coefficients are 0.99, 0.94 and 0.98 for MA, DMA and TMA, respectively".

Some of less important items:

Pg. 3, line 41: What are "anthropological" emissions?

**Response**: Thank you. The word will be replaced with "anthropogenic".

Pg. 6, line 138: Tabor et al., 1994, Ullerstam et al., 2003: references are missing.

**Response**: Thank you so much. These references will be added.

Pg. 13, 295: Salgado-Muñoz and,,,,also in bibliographic list at the end (line 623)!

**Response**: Thank you. This will be corrected as "(Salgado-Muñoz and Rossi, 2002)"

Pg. 14, line 327: cluster has the wrong polarity!

**Response**: We double checked this cluster in the reference (Bzdek et al., 2010). It was $[(NH_4)_3(HSO_4)_2]^+$ instead of $[(NH_4)_3(SO_4)_2]^+$. This will be corrected in the manuscript.

Pg. 14, line 332: Larger than what? The author's comparison only has one leg!!

**Response**: Thank you. This sentence will be revised as "$H_2SO_4$ solution with high $H_2SO_4$ content also showed a slightly larger $\gamma$ of amines than that with low $H_2SO_4$ concentration".

Pg. 15, line 344: What is TEAH sulfate?

**Response**: It should be TEA-sulfate. It will be corrected in the manuscript.

Pg. 20, line 473: mineral dust is a bad reservoir or no reservoir at all! The authors should cut out qualitative or meaningless talk.

**Response**: Thank you. The last sentence "In this process, mineral dust takes on the role of a carrier or reservoir of amines" will be deleted in the manuscript.

Pg. 23, line 653: "Physiochemical"?

**Response**: Yes, it is. The title of this paper is:

[Figure]

Article

pubs.acs.org/est

**Physiochemical Properties of Alkylaminium Sulfates: Hygroscopicity, Thermostability, and Density**

Chong Qiu[†] and Renyi Zhang[*,†,‡]

[†]Department of Chemistry and [‡]Department of Atmospheric Sciences, Texas A&M University, College Station, Texas 77843, United States

Pg. 25, Table 1: Under DMA: fifth entry from the top of DMA field has wrong polarity!

**Response**: Thank you. It should be "$[(NH_4)_3(HSO_4)_2]^+$" and will be corrected in the manuscript.

**References:**

Auerbach, S. M., Carrado, K. A., and Dutta, P. K.: Handbook of Zeolite Science and Technology, CRC Press, New York, 443 pp., 2003.

Beichert, P., and Finlayson-Pitts, B. J.: Knudsen cell studies of the uptake of gaseous HNO3 and other oxides of nitrogen on solid NaCl: the role of surface-adsorbed water. , J. Phys. Chem., 100, 15218-15228, 1996.

Bzdek, B. R., Ridge, D. P., and Johnston, M. V.: Amine exchange into ammonium bisulfate and ammonium nitrate nuclei, Atmos. Chem. Phys., 10, 3495-3503, doi: 10.5194/acp-10-3495-2010, 2010.

Compton, R. G., Bamford, C. H., and Tipper, C. F. H.: Simple Processes at the Gas-Solid Interface, 1st ed., Elsevier Science Publishing Company Inc., 193 pp., 1984.

De Moor, B. A., Reyniers, M.-F., Sierka, M., Sauer, J., and Marin, G. B.: Physisorption and Chemisorption of Hydrocarbons in H-FAU Using QM-Pot(MP2//B3LYP) Calculations, The Journal of Physical Chemistry C, 112, 11796-11812, 10.1021/jp711109m, 2008.

Fan, T.-B., and Wang, L.-S.: A viscosity model based on Peng–Robinson equation of state for light hydrocarbon liquids and gases, Fluid Phase Equilibria, 247, 59-69, http://dx.doi.org/10.1016/j.fluid.2006.06.008, 2006.

Fenter, F. F., Caloz, F., and Rossi, M. J.: Kinetics of Nitric Acid Uptake by Salt, J. Phys. Chem., 98, 9801-9810, 1994.

Freshour, N. A., Carlson, K. K., Melka, Y. A., Hinz, S., Panta, B., and Hanson, D. R.: Amine permeation sources characterized with acid neutralization and sensitivities of an amine mass spectrometer, Atmos. Meas. Tech., 7, 3611-3621, 10.5194/amt-7-3611-2014, 2014.

Griffiths, P. T., and Anthony Cox, R.: Temperature dependence of heterogeneous uptake of N2O5 by ammonium sulfate aerosol, Atmos. Sci. Lett., 10, 159-163, doi: 10.1002/asl.225, 2009.

Hallquist, M., Stewart, D. J., Jacob Baker, A., and Cox, R. A.: Hydrolysis of N2O5 on Submicron Sulfuric Acid Aerosols, J. Phys. Chem. A, 104, 3984-3990, doi: 10.1021/jp9939625, 2000.

Hoffman, R. C., Kaleuati, M. A., and Finlayson-Pitts, B. J.: Knudsen Cell Studies of the Reaction of Gaseous HNO3 with NaCl Using Less than a Single Layer of Particles at 298 K: A Modified Mechanism, J. Phys. Chem. A., 107, 7818-7826, doi: 10.1021/jp030611o, 2003.

Lin, J.-L., Lin, Y.-C., Lin, B.-C., Lai, P.-C., Chien, T.-E., Li, S.-H., and Lin, Y.-F.: Adsorption and

Reactions on TiO2: Comparison of N,N-Dimethylformamide and Dimethylamine, J. Phys. Chem. C, 118, 20291-20297, doi: 10.1021/jp5044859, 2014.

Liu, Y., He, H., and Mu, Y.: Heterogeneous reactivity of carbonyl sulfide on $\alpha$-$Al_2O_3$ and $\gamma$-$Al_2O_3$, Atmos. Environ., 42, 960-969, doi: 10.1016/j.atmosenv.2007.10.007, 2008.

Liu, Y., Ma, J., and He, H.: Heterogeneous reactions of carbonyl sulfide on mineral oxides: mechanism and kinetics study, Atmos. Chem. Phys., 10, 10335-10344, doi: 10.5194/acp-10-10335-2010, 2010a.

Liu, Y., Ma, J., Liu, C., and He, H.: Heterogeneous uptake of carbonyl sulfide onto kaolinite within a temperature range of 220–330 K, J. Geophys. Res., 115(D24311), doi:10.1029/2010JD014778, 2010b.

Liu, Y., Ma, Q., and He, H.: Heterogeneous uptake of amines by citric acid and humid acid, Environ. Sci Technol., 46, 11112-11118, doi: 10.1021/es302414v, 2012.

Ma, Q., Liu, Y., Liu, C., Ma, J., and He, H.: A case study of Asian dust storm particles: Chemical composition, reactivity to SO2 and hygroscopic properties, J. Environ. Sci., 24, 62-71, 2012.

Matthews, G. P., Ridgway, C. J., and Small, J. S.: Modelling of simulated clay precipitation within reservoir sandstones, Murine and Petroleum Geology, 13, 581-589, 1996.

Miranda-Trevino, J. C., and Coles, C. A.: Kaolinite properties, structure and influence of metal retention on pH, Appl. Clay Sci., 23, 133-139, doi:10.1016/S0169-1317(03)00095-4, 2003.

Neuman, J. A., Ryerson, T. B., Huey, L. G., Jakoubek, R., Nowak, J. B., Simons, C., and Fehsenfeld, F. C.: Calibration and Evaluation of Nitric Acid and Ammonia Permeation Tubes by UV Optical Absorption, Environ. Sci. Technol., 37, 2975-2981, 10.1021/es026422l, 2003.

Nunes, M. H. O., Silva, V. T. d., and Schmal, M.: The effect of copper loading on the acidity of Cu/HZSM-5 catalysts: IR of ammonia and methanol for methylamines synthesis, Appl. Catal. A: General, 294, 148-155, doi: 10.1016/j.apcata.2005.06.031, 2005.

Ongwandee, M., Morrison, G. C., Guo, X., and Chusuei, C. C.: Adsorption of trimethylamine on zirconium silicate and polyethylene powder surfaces, Colloid Surf. A: Physicochem. Eng. Asp., 310, 62-67, doi: 10.1016/j.colsurfa.2007.05.076, 2007.

Salgado-Muñoz, M. S., and Rossi, M. J.: Heterogeneous reactions of HNO3 with flame soot generated under different combustion conditions. Reaction mechanism and kinetics, Phys. Chem. Chem. Phys., 4, 5110-5118, doi: 10.1039/b203912p, 2002.

Seisel, S., Lian, Y., Keil, T., Trukhin, M. E., and Zellner, R.: Kinetics of the interaction of water vapour with mineral dust and soot surfaces at T = 298 K, Phys. Chem. Chem. Phys., 6, 1926-1932, doi: 10.1039/B314568A, 2004.

Seisel, S., Börensen, C., Vogt, R., and Zellner, R.: Kinetics and mechanism of the uptake of $N_2O_5$ on mineral dust at 298K, Atmos. Chem. Phys., 5, 3423-3432, doi: 10.5194/acp-5-3423-2005, 2005.

Tunon, I., Silla, E., and Tomasi, J.: Methylamines basicity calculations: in vacuo and in solution comparative analysis, J. Phys. Chem., 96, 9043-9048, 10.1021/j100201a065, 1992.

Ullerstam, M., Johnson, M. S., Vogt, R., and Ljungström, E.: DRIFTS and Knudsen cell study of the heterogeneous reactivity of SO2 and NO2 on mineral dust. . Atmos. Chem. Phys., 3 2043-2051, 2003.

Underwood, G. M., Li, P., Usher, C. R., and Grassian, V. H.: Determining accurate kinetic parameters of potentially important heterogeneous atmospheric reaction on solid particle with a Knudsen cell reactor, J. Phys. Chem. A, 104 819-829, 2000.

Underwood, G. M., Li, P., Al-Abadleh, H. A., and Grassian, V. H.: A Knudsen cell study of the heterogeneous reactivity of nitric acid on oxide and mineral dust particles, J. Phys. Chem. A, 105, 6609-6620, doi: 10.1021/jp002223h, 2001.

Wardhana, Y. W., Hasanah, A. N., and Primandini, P.: Deformation and adsorption capacity of kaolin that

is influenced by temperature variation on calcination, Int. J. Pharm. Pharm. Sci., 6,Suppl 3, 1-2, 2014.

---

## Author Comment (AC2) · 22 Sep 2016

**Referee #2**

Using a Knudsen cell reactor and ATR-FTIR, Liu et al. investigated the heterogeneous reactions of methylamine (MA), dimethylamine (DMA), and trimethylamine (TMA) with kaolinite (as a surrogate of mineral dust) and the effect of temperature. Both amines and mineral dust are important components in the troposphere, and their reactions have not been examined yet. This manuscript fits the scope of ACP well and the results are quite new. The kinetic data present by this work would help us better understand the tropospheric sinks of amines and the aging processes of mineral dust particles. This manuscript can be published after the following comments are addressed
**Response:** Thank you for your positive comments.

**Major comments:**

Line 133-136: Prior to the uptake measurement, the reaction chamber was passivated with amines to reduce/minimize the wall effect. The sample chamber also has some (though smaller compared to the reaction chamber) wall effect. Is this significant compared to the uptake by kaolinite? I believe this can be determined by background experiments in which no dust is deposited onto the sample holder.

**Response**: Thank you for your comment. We have performed blank experiments for all of these amines. Figure R1 shows the results (raw data) of the blank experiments for MA and TMA. This experiment was carried out at 232 K. When the steady state of TMA signals was reached (both m/z 59 and 58), the Teflon sampler (without particles) was exposed to TMA vapor by opening the sample lid. No changes in TMA signals were observed. This means the contribution of the fresh surface to the uptake of amines

by kaolinite is negligible.

[Figure]

**Fig. R1**. Blank experiments (raw data) for (A) MA and (B) TMA at 232 K.

Line 429-438: While I agree with the authors that heterogeneous reactions with mineral dust can be an important sink for these amines in the troposphere, I also two comments: 1) the effect of gas phase diffusion needs to be discussed (Tang et al., 2015), especially for large particles (e.g., mineral dust) and fast uptake (which is also the case in this study); 2) only extreme conditions with very high dust loadings are discussed; to understand the general role of these reactions, the authors should also discuss the lifetimes under typical atmospheric conditions. By the way, the dust loading unit used in this manuscript is $\mu m^2\ cm^{-3}$; while this is convenient to calculate the lifetime using Eq. (16), the corresponding mass concentration (which is more widely used) should also be provided.

**Response**: Thank you for your suggestion. We agree with you that gas-phase diffusion would be an important issue if the uptake experiments were not performed in a free molecular flow. Thus, a diffusion correction was required using the empirical formula for experiments carried out at ambient pressure (Fuchs and Sutugin, 1970;Worsnop et

al., 2002;Widmann and Davis, 1997). However, the pressure in the reactor was $\sim3.5\times10^{-4}$ Torr in this study. Thus, the mean free path of gas was 12 cm even at 232 K. This value was greater than the reactor dimensions (5.6 cm), which will be summarized in **Table S1** (Table R1, here). Therefore, a diffusion correction was not necessary in this study when calculating the effective uptake coefficient.

**Table R1**. The parameters of the Knudsen cell reactor.

| Parameter | value |
|---|---|
| $V$ (cm$^3$) | 93.90 |
| $A_w$ (cm$^2$) | 116.20 |
| $\Phi$(cm) | 5.60 |
| $H$(cm) | 3.50 |
| $A_s$ (cm$^2$) | 3.26 |
| $A_h$ (N$_2$, mm$^2$) | $0.90\pm0.15$ |
| $k_{esc}$ (N$_2$, s$^{-1}$) | $1.14\pm0.14$ |
| $F$ of amines (molecules s$^{-1}$) | $\sim1.2\times10^{13}$ |
| MS sensitivity of amines ($3\sigma$, molecules s$^{-1}$) | $\sim1.1\times10^{12}$ (MA), $\sim1.4\times10^{12}$ (DMA), $\sim1.7\times10^{12}$ (TMA) |
| $D$p of kaolinite ($\mu$m) | 0.56 |

Of course, diffusion should be considered in modelling studies to assess the impact of heterogeneous uptake on the sink of amines. Because the diffusion effect depends on the size distribution of particles, it cannot be estimated in Eq. (16) at the present time. Using Eq. (16), we can only roughly estimate the lifetimes of amines via heterogeneous uptake. In the manuscript, the following paragraph will be added to clarify the influence on diffusion of lifetime estimation "It should be pointed out that the diffusion effect of amines from the gas phase to the particle surface at ambient pressure will decrease the apparent loss rate of amines (Tang et al., 2015), and consequently enhance their lifetimes in the atmosphere. This was not considered in Eq. (16). Therefore, modelling studies considering this influence are necessary in the future for fully understanding the impacts of heterogeneous uptake on the sink of amines".

With the average density (1.5 g cm$^{-3}$) of atmospheric particles (Kannosto et al., 2008), the surface concentration of particles can be converted to the mass concentration;

150 μm$^2$ cm$^{-3}$ corresponds to 259 μg·cm$^{-3}$. This will be added in the manuscript as "If we assume that all mineral dust is in the form of kaolinite, and the dust loading is 150 μm$^2$ cm$^{-3}$ (de Reus et al., 2000; Frinak et al., 2004), which corresponds to 259 μg·cm$^{-3}$ with an average density of atmospheric particles of 1.5 g cm$^{-3}$ (Kannosto et al., 2008), under extreme conditions". The lifetimes of amines by heterogeneous uptake under a typical particle concentration (80 μg·cm$^{-3}$) (He et al., 2014) have also been estimated. Their lifetimes were 49.4, 56.6 and 18.6 h, respectively. This will also be added in the manuscript as "Under the typical particle concentration in Beijing (80 μg·cm$^{-3}$) (He et al., 2014), their lifetimes were 49.4, 56.6 and 18.6 h, respectively" and "Under the typical particle concentration in Beijing, the contribution of heterogeneous uptake to the sink of amines should also be considered".

Line 467-468: It is stated that physical adsorption takes place between amines and kaolinites, but no direct experimental evidence is provided. As I understand, both Knudsen cell reactor and ATR-FTIR can be used to examine whethere a gas-surface reaction is reversible. I would suggest that another 1-2 figures with experimental data should be included to clarify this issue.

**Response**: Thank you for your suggestion. Repeat uptake experiments were carried out after the used sample was evacuated overnight at 300 K. The uptake curve in the second run coincided very well with that in the first run (Fig. R2). This means that uptake of amines on kaolinite is reversible. This figure will be added as Fig. S4.

[Figure]

**Fig. R2.** Repeat uptake curves of MA after the used sample was evacuated overnight at 300 K. The sample mass was 19.8 mg.

**Minor comments:**

Line 4: I should suggest that "amines" should be changed to "methylamine, dimethylamine, and trimethylamine (TMA)" to be specific.

**Response**: Thank you. The "amines" in the title will be replaced with "methylamine, dimethylamine, and trimethylamine" in the manuscript.

Line 24-25: This statement is incorrect. The uptake coefficients were directly derived from the experimental data as discussed in Sections 3.1 and 3.2, and mass accommodation coefficients are used to derive enthalpies and entropies (Section 3.3).

**Response**: Thank you. This sentence will be revised as "The uptake coefficients ($\gamma$) were mainly determined by mass accommodation coefficients based on the temperature dependence of the $\gamma$" in the manuscript.

Line 49: please also provide the concentrations in pptv.

**Response**: Thank you. The stated concentration range corresponds to 20−340 pptv.

This will be added as "Amines, whose atmospheric concentrations are typically 1~14 nmol N m$^{-3}$ (or 20−340 pptv)…" in the manuscript.

Line 81: The review paper by Crowley et al. (2010) should also be cited here together with Usher et al. (2003).

**Response**: These two references will be cited in the manuscript.

Line 312: I believe "$\gamma$eff" should be "$\gamma$eff/$\gamma$obs".

**Response**: Thank you. It will be corrected in the manuscript.

Line 339-343: It should be further explained why the study by Wang et al. (2010a) explained the difference between Liu et al. (2012a) and Qiu et al. (2011). For the current manuscript, it is not clear to me.

**Response**: The temperature was 298 K in our previous work, while it was 293 K in Qiu's work. This was partially responsible for the smaller uptake coefficient in our work. We will added this reason in the manuscript as "This might partially explain the difference in the measured $\gamma$ of MA on $(NH_4)_2SO_4$ between our previous work (Liu et al., 2012) and Qiu's work (Qiu et al., 2011) because the temperature was 5 K lower in their work than ours".

Line 457-463: the effects of heterogeneous reactions on the chemical composition and IN activity of mineral dust particles is mentioned here. I do believe that it should also be mentioned in the introduction. Besides, many papers have discussed the effects of heterogeneous reaction on the hygroscopicity and CCN and IN activities of mineral dust, including those by Cziczo et al. (2009), Sullivan et al. (2009), Ma et al. (2012), Tobo et al. (2012) and Tang et al. (2016), just to name a few.

**Response**: Thank you for your suggestion. A paragraph will be added in the manuscript as "In addition, mineral dust including kaolinite has been well recognized as effective ice nuclei (IN) (Wex et al., 2014;Salam et al., 2006). Surface coatings from heterogeneous reaction may modify the hygroscopicity (Ma et al., 2012), the cloud

condensation nuclei (CCN) activity (Sullivan et al., 2009) and the ice nuclei (IN) activity (Cziczo et al., 2009;Tobo et al., 2012) of mineral dust. Thus, it is necessary to investigate the heterogeneous reaction between amines and kaolinite for understanding the climatic effect changes of kaolinite during atmospheric transformation."

Figure 5: It will improve the readability of this figure to move $\Delta H$ and $\Delta S$ values to the figure caption instead.

**Response**: Thank you. These values will be moved to the figure caption as "The $\Delta H$ of MA, DMA and TMA are -11.2±0.6, -15.8±3.4 and -12.1±1.2 kJ mol$^{-1}$, while the $\Delta S$ are -97.5±2.4, -111.8±13.0 and -90.4±4.9 J mol$^{-1}$K$^{-1}$, respectively."

**References:**
Cziczo, D. J., Froyd, K. D., Gallavardin, S. J., Moehler, O., Benz, S., Saathoff, H., and Murphy, D. M.: Deactivation of ice nuclei due to atmospherically relevant surface coatings, Environ. Res. Lett., , 4, doi:10.1088/1748-9326/4/4/044013, 2009.

Fuchs, N. A., and Sutugin, A. G.: Highly Dispersed Aerosols, Butterworth-Heinemann, Newton, MA, 1970.

He, H., Wang, Y., Ma, Q., Ma, J., Chu, B., Ji, D., Tang, G., Liu, C., Zhang, H., and Hao, J.: Mineral dust and NOx promote the conversion of SO2 to sulfate in heavy pollution days, Sci. Rep., 4, 10.1038/srep04172, 2014.

Kannosto, J., Virtanen, A., Lemmetty, M., Mäkelä, J. M., Keskinen, J., Junninen, H., Hussein, T., Aalto, P., and Kulmala, M.: Mode resolved density of atmospheric aerosol particles, Atmos. Chem. Phys., 8, 5327-5337, 10.5194/acp-8-5327-2008, 2008.

Liu, Y., Han, C., Liu, C., Ma, J., Ma, Q., and He, H.: Differences in the reactivity of ammonium salts with methylamine, Atmos. Chem. Phys., 12, 4855-4865, doi: 10.5194/acp-12-4855-2012, 2012.

Ma, Q., Liu, Y., Liu, C., and He, H.: Heterogeneous reaction of acetic acid on MgO, $\alpha$-Al$_2$O$_3$, and CaCO$_3$ and the effect on the hygroscopic behaviour of these particles, Phys. Chem. Chem. Phys., 14, 8403-8409, 2012.

Qiu, C., Wang, L., Lal, V., Khalizov, A. F., and Zhang, R.: Heterogeneous Reactions of Alkylamines with Ammonium Sulfate and Ammonium Bisulfate, Environ. Sci. Technol., 45, 4748-4755, doi: 10.1021/es1043112, 2011.

Salam, A., Lohmann, U., Crenna, B., Lesins, G., Klages, P., Rogers, D., Irani, R., MacGillivray, A., and Coffin, M.: Ice Nucleation Studies of Mineral Dust Particles with a New Continuous Flow Diffusion Chamber, Aerosol Sci.Technol., 40, 134-143, doi: 10.1080/02786820500444853, 2006.

Sullivan, R. C., Moore, M. J. K., Petters, M. D., Kreidenweis, S. M., Roberts, G. C., and Prather, K. A.: Timescale for hygroscopic conversion of calcite mineral particles through heterogeneous reaction with nitric acid, Phys. Chem. Chem. Phys., 11, 7826-7837, 10.1039/B904217B, 2009.

Tang, M. J., Shiraiwa, M., Pöschl, U., Cox, R. A., and Kalberer, M.: Compilation and evaluation of gas phase diffusion coefficients of reactive trace gases in the atmosphere: Volume 2. Diffusivities of organic

compounds, pressure-normalised mean free paths, and average Knudsen numbers for gas uptake calculations, Atmos. Chem. Phys., 15, 5585-5598, 10.5194/acp-15-5585-2015, 2015.

Tobo, Y., DeMott, P. J., Raddatz, M., Niedermeier, D., Hartmann, S., Kreidenweis, S. M., Stratmann, F., and Wex, H.: Impacts of chemical reactivity on ice nucleation of kaolinite particles: A case study of levoglucosan and sulfuric acid, Geophys. Res. Lett., 39, doi: 19810.11029/12012gl053007, 2012.

Wex, H., DeMott, P. J., Tobo, Y., Hartmann, S., Rösch, M., Clauss, T., Tomsche, L., Niedermeier, D., and Stratmann, F.: Kaolinite particles as ice nuclei: learning from the use of different kaolinite samples and different coatings, Atmos. Chem. Phys., 14, 5529-5546, doi: 10.5194/acp-14-5529-2014, 2014.

Widmann, J. F., and Davis, E. J.: Mathematical models of the uptake of ClONO2 and other gases by atmospheric aerosols, J. Aerosol Sci., 28, 87-106, doi: 10.1016/S0021-8502(96)00060-2, 1997.

Worsnop, D. R., Morris, J. W., Shi, Q., Davidovits, P., and Kolb, C. E.: A chemical kinetic model for reactive transformations of aerosol particles, Geophys. Res. Lett., 29, 1996, doi: 10.1029/2002gl015542, 2002.

---

## Editor Comment (EC1) · T. Bartels-Rausch (Editor) · 18 Oct 2016

Dear Dr. Liu

Thank you again for submitting your manuscript "Heterogeneous uptake of amines onto kaolinite in the temperature range of 232-300 K" to ACP.

After thorough consideration and detailed discussion with one of the referees, I decided to revoke my initial decision of accepting your manuscript for publication. Instead, I will send the manuscript out for an additional round of peer review. This is certainly an unusual measure and will increase the time that your manuscript is under review and will require you to reply to additional referee comments. This is in light of an even more unfavourable alternative in which the discussion would continue via comments on the published manuscript in ACP.

[Figure]

After my initial decision, one of the referees contacted me raising strong issues and urgently asked for further changes to the manuscript prior to publication. It turned out, that we do not fully agree on the judgement on how satisfactory the first referee comments were implemented in the revised manuscript. In light of the new arguments, I'm convinced that it suits ACP better to discuss the referee's concerns publicly in ACPD. This does not only give you the opportunity to directly reply to the comments, but independent judgement by additional referee can be considered. Furthermore, changes to the manuscript might be made as considered necessary after the second round of peer review. Changing my decision rather than overruling the referee's concerns, will hopefully serve to foster a public discussion within the science community on the very fundamental remarks about the basics of the Knudsen technique as well as on the specific issues raised by the referee.

I apologise for the confusion that this might have caused and for any trouble that this certainly has caused; and hope that you are willing to continue the discussion.

Kind regards, Thorsten Bartels-Rausch

―――――――――――――――――――